# Efficient Φ-Regret Minimization with Low-Degree Swap Deviations in Extensive-Form Games

**Brian Hu Zhang**
Carnegie Mellon University
bhzhang@cs.cmu.edu

**Ioannis Anagnostides**
Carnegie Mellon University
ianagnos@cs.cmu.edu

**Gabriele Farina**
MIT
gfarina@mit.edu

**Tuomas Sandholm**
Carnegie Mellon University
Strategic Machine, Inc.
Strategy Robot, Inc.
Optimized Markets, Inc.
sandholm@cs.cmu.edu

## Abstract

Recent breakthrough results by Dagan, Daskalakis, Fishelson and Golowich [2023] and Peng and Rubinstein [2023] established an efficient algorithm attaining at most $\epsilon$ *swap regret* over extensive-form strategy spaces of dimension $N$ in $N^{\tilde{O}(1/\epsilon)}$ rounds. On the other extreme, Farina and Pipis [2023] developed an efficient algorithm for minimizing the weaker notion of *linear-swap* regret in $\text{poly}(N)/\epsilon^2$ rounds. In this paper, we develop efficient parameterized algorithms for regimes between these two extremes. We introduce the set of $k$-*mediator deviations*, which generalize the *untimed communication deviations* recently introduced by Zhang, Farina and Sandholm [2024] to the case of having multiple mediators, and we develop algorithms for minimizing the regret with respect to this set of deviations in $N^{O(k)}/\epsilon^2$ rounds. Moreover, by relating $k$-mediator deviations to low-degree polynomials, we show that regret minimization against degree-$k$ polynomial swap deviations is achievable in $N^{O(kd)^3}/\epsilon^2$ rounds, where $d$ is the depth of the game, assuming a constant branching factor. For a fixed degree $k$, this is polynomial for Bayesian games and quasipolynomial more broadly when $d = \text{polylog} N$—the usual balancedness assumption on the game tree. The first key ingredient in our approach is a relaxation of the usual notion of a fixed point required in the framework of Gordon, Greenwald and Marks [2008]. Namely, for a given deviation $\phi$, we show that it suffices to compute what we refer to as a *fixed point in expectation*; that is, a distribution $\pi$ such that $\mathbb{E}_{\boldsymbol{x} \sim \pi}[\phi(\boldsymbol{x}) - \boldsymbol{x}] \approx 0$. Unlike the problem of computing an actual (approximate) fixed point $\boldsymbol{x} \approx \phi(\boldsymbol{x})$, which we show is PPAD-hard, there is a simple and efficient algorithm for finding a solution that satisfies our relaxed notion. As a byproduct, we provide, to our knowledge, the fastest algorithm for computing $\epsilon$-correlated equilibria in normal-form games in the medium-precision regime, obviating the need to solve a linear system in every round. Our second main contribution is a characterization of the set of low-degree deviations, made possible through a connection to low-depth decisions trees from Boolean analysis.

38th Conference on Neural Information Processing Systems (NeurIPS 2024).

# 1   Introduction

*Correlated equilibrium (CE)*, introduced in a groundbreaking work by Aumann [1974], has emerged as one of the most influential solution concepts in game theory. Often contrasted with *Nash equilibrium* [Nash, 1950], it is regarded by many as more natural; in the words attributed to another Nobel laureate, Roger Myerson, "if there is intelligent life on other planets, in a majority of them, they would have discovered correlated equilibrium before Nash equilibrium." Correlated equilibria also enjoy more favorable computational properties: unlike Nash equilibria, they can be expressed as solutions to a linear program, thereby enabling their computation in polynomial time, at least in *normal-form* games [Papadimitriou and Roughgarden, 2008, Jiang and Leyton-Brown, 2011]. Further, a correlated equilibrium arises through repeated play from natural *no-regret* learning dynamics [Hart and Mas-Colell, 2000, Foster and Vohra, 1997].

However, many real-world strategic interactions feature sequential moves and imperfect information. In such scenarios, the so-called *extensive form* constitutes the canonical game representation [Kuhn, 1953, Shoham and Leyton-Brown, 2009]: a normal-form description of the game would be prohibitively large. It is startling to realize that 50 years after Aumann's original work, the complexity of computing correlated equilibria in extensive-form games—sometimes referred to as *normal-form correlated equilibria (NFCE)* to disambiguate from other pertinent but weaker solution concepts—remains an outstanding open problem [von Stengel and Forges, 2008, Papadimitriou and Roughgarden, 2008].

The long-standing absence of efficient algorithms for computing an NFCE shifted the focus to natural relaxations thereof, which can be understood through the notion of $\Phi$-*regret* [Greenwald and Hall, 2003, Stoltz and Lugosi, 2007, Rakhlin et al., 2011]. In particular, $\Phi$ represents a set of strategy deviations; the richer the set of deviations, the stronger the induced solution concept. When $\Phi$ contains all possible transformations, one recovers the notion of NFCE—corresponding to *swap regret*. At the other end of the spectrum, *coarse correlated equilibria* correspond to $\Phi$ consisting solely of constant transformations (aka. *external regret*). Perhaps the most notable relaxation is the *extensive-form correlated equilibrium (EFCE)* [von Stengel and Forges, 2008], which can be computed exactly in time polynomial in the representation of the game tree [Huang and von Stengel, 2008]. Considerable interest in the literature has recently been on *learning dynamics* that minimize $\Phi$-regret (*e.g.*, Morrill et al. [2021a,b], Bai et al. [2022], Bernasconi et al. [2023], Noarov et al. [2023], Dudík and Gordon [2009], Gordon et al. [2008], Fujii [2023], Dann et al. [2023], Mansour et al. [2022]). A key reference point in this line of work is the recent construction of Farina and Pipis [2023], an efficient algorithm minimizing *linear swap regret*—that is, the notion of $\Phi$-regret where $\Phi$ contains all *linear* deviations. Such algorithms lead to an $\epsilon$-equilibrium in time polynomial in the game's description and $1/\epsilon$—aka. a fully polynomial-time approximation scheme (FPTAS).

Yet, virtually nothing was known beyond those special cases until recent breakthrough results by Dagan et al. [2024] and Peng and Rubinstein [2024], who introduced a new approach for reducing swap regret to external regret; unlike earlier reductions [Gordon et al., 2008, Blum and Mansour, 2007, Stoltz and Lugosi, 2005], their algorithm can be implemented efficiently even in certain settings with an exponential number of pure strategies. For extensive-form games, their reduction implies a polynomial-time approximation scheme (PTAS) for computing an $\epsilon$-correlated equilibrium; their algorithm has complexity $N^{\tilde{O}(1/\epsilon)}$ for games of size $N$, which is polynomial only when $\epsilon$ is an absolute constant. Unfortunately, it was thereafter shown that in the usual regime of interest, where instead $\epsilon \le \mathsf{poly}(1/N)$, an exponential number of rounds is inevitable even against an oblivious adversary [Daskalakis et al., 2024]. In light of that lower bound, our focus here is on developing algorithms attaining a better complexity bound of $\mathsf{poly}(N, 1/\epsilon)$—the typical guarantee one hopes for within the no-regret framework—by considering a more structured but rich class of deviations $\Phi$.

# 2   Preliminaries

Before we proceed by giving an overview of our results and technical contributions, we first introduce some basic background on tree-form decisions problems and $\Phi$-regret minimization.

## 2.1 Tree-form decision problems

A *tree-form decision problem* describes a sequential interaction between a *player* and a (possibly adversarial) *environment*. There is a tree of *nodes*. The root is denoted $\varnothing$. We will use $s \in \mathcal{S}$ to denote a generic node, and $p_s$ (where $s \neq \varnothing$) to denote the parent of $s$. Leaves are called *terminal nodes*; a generic terminal node is denoted $z \in \mathcal{Z}$. Internal nodes can be one of three types: *decision points*, where the player plays an action, *observation points*, where the environment picks the next decision point. A generic decision point will be denoted $j$, and the set of actions at $j$ will be denoted $\mathcal{A}_j$. The child node reached by following action $a \in \mathcal{A}_j$ is denoted $ja$. We will use $N$ to denote the number of terminal nodes. We will also assume without loss of generality that all decision points have branching factor at least 2, and that decision and observation points alternate. Thus, the total number of nodes in the tree is also $O(N)$. The *depth* of a decision problem is the largest number of decision points in any root-to-terminal-node path. An example of a tree-form decision problem is depicted below in Figure 1.

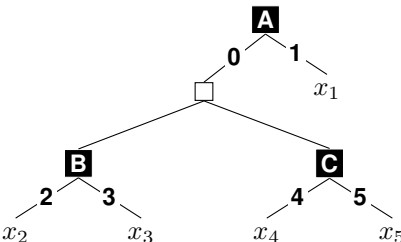

Figure 1: An example of a tree-form decision problem. Decision points are black squares with white text labels; observataion points are white squares. Edges are labeled with action names, which are numbers. Pure strategies in this decision problem are identified with vectors $\boldsymbol{x} = (x_1, x_2, x_3, x_4, x_5) \in \{0,1\}^5$ satisfying $1 - x_1 = x_2 + x_3 = x_4 + x_5$.

A *pure strategy* consists of an assignment of one action $a_j \in \mathcal{A}_j$ to each decision point $j$. The *tree-form representation* of the pure strategy is the vector $\boldsymbol{x} \in \{0,1\}^N$ where $\boldsymbol{x}[z] = 1$ if and only if the player plays all the actions on the $\varnothing \to z$ path. Although $\boldsymbol{x}$ is a vector indexed only by terminal nodes, we also overload notation to write $\boldsymbol{x}[s] = 1$ if and only if the player plays all actions on the $\varnothing \to s$ path (In other words, $\boldsymbol{x}[s] = 1$ if there exists some $z \succeq s$ with $\boldsymbol{x}[z] = 1$). Multiple pure strategies can have the same tree-form representation, but in this paper we will only concern ourselves with strategies in tree-form representation, and thus for our purposes such strategies will be treated as identical. We will use $\mathcal{X} \subseteq \{0,1\}^N$ to denote the set of tree-form strategies, and sometimes (when context is clear) we will also use $\mathcal{X}$ to denote the tree-form decision problem itself. For a point in the convex hull of $\mathcal{X}$, $\operatorname{conv} \mathcal{X}$, we also use the symbol $\boldsymbol{x} \in \operatorname{conv} \mathcal{X}$. For *mixed* strategies, we instead use $\pi \in \Delta(\mathcal{X})$. When it is relevant, we assume that utilities are rational numbers representable with $\operatorname{poly}(N)$ bits.

## 2.2 Regret minimization

In the framework of online learning, a learner interacts with an adversary over a sequence of rounds. In each round, the learner selects a strategy, whereupon the adversary constructs a utility function. Throughout this paper, we operate in the *full feedback* setting, wherein the learner gets to observe the entire utility function produced by the adversary after each round. We allow the adversary to be *strongly adaptive*, so that the (linear) utility function at the $t$th round $u^{(t)} : \mathcal{X} \ni \boldsymbol{x} \mapsto \langle \boldsymbol{u}^{(t)}, \boldsymbol{x} \rangle$ can depend on the strategy of the learner at that round; this is a standard assumption (*cf.* the notion of *leaky forecasts* in the context of calibration [Foster and Hart, 2018]) that will be used for our lower bound (Theorem 3.3). We assume that utilities belong to $\mathcal{U} := \{\boldsymbol{u} : |\langle \boldsymbol{u}, \boldsymbol{x} \rangle| \leq 1, \forall \boldsymbol{x} \in \mathcal{X}\}$. It will be convenient to use $\|\boldsymbol{x}\|_{\mathcal{X}} := \max_{\boldsymbol{u} \in \mathcal{U}} \langle \boldsymbol{u}, \boldsymbol{x} \rangle$ for the induced norm.

We measure the performance of an online learning algorithm as follows. Suppose that $\Phi \subseteq (\operatorname{conv} \mathcal{X})^{\mathcal{X}}$ is a set of deviations. If the learner outputs in each round a *mixed strategy* $\pi^{(t)} \in \Delta(\mathcal{X})$,

its (time-average) $\Phi$-*regret* [Greenwald and Hall, 2003, Stoltz and Lugosi, 2007] is defined as

$$\overline{\text{Reg}}_\Phi^T := \frac{1}{T} \max_{\phi \in \Phi} \sum_{t=1}^T \left\langle \boldsymbol{u}^{(t)}, \mathop{\mathbb{E}}_{\boldsymbol{x}^{(t)} \sim \pi^{(t)}} [\phi(\boldsymbol{x}^{(t)}) - \boldsymbol{x}^{(t)}] \right\rangle. \tag{1}$$

In the special case where $\Phi$ contains only *constant transformations*, one recovers the notion of *external regret*. On the other extreme, *swap regret* corresponds to $\Phi$ containing all functions $\mathcal{X} \to \mathcal{X}$.

It is sometimes assumed that the learner instead selects in each round a strategy $\boldsymbol{x}^{(t)} \in \text{conv}\,\mathcal{X}$. To translate (1) in that case, we introduce the *extended mapping* of a deviation $\phi : \mathcal{X} \to \text{conv}\,\mathcal{X}$ as $\phi^\delta := \mathbb{E}_{\boldsymbol{x}' \sim \delta(\boldsymbol{x})}[\phi(\boldsymbol{x}')]$, where $\delta : \text{conv}\,\mathcal{X} \to \Delta(\mathcal{X})$ is a function that is *consistent* in the sense that $\mathbb{E}_{\boldsymbol{x}' \sim \delta(\boldsymbol{x})}[\boldsymbol{x}'] = \boldsymbol{x}$. A canonical example of such a function $\delta$ is the *behavioral strategy map* $\beta : \text{conv}\,\mathcal{X} \to \Delta(\mathcal{X})$, which returns the unique (ignoring actions at decision points reached with probability zero) mixed strategy whose actions at different decision points are independent and whose expectation is $\boldsymbol{x}$. We give another example of a consistent map later in Appendix C.2. Accordingly, we let $\Phi^\delta$ denote all extended mappings. In this context, $\Phi^\delta$-regret is defined as

$$\overline{\text{Reg}}_{\Phi^\delta}^T := \frac{1}{T} \max_{\phi^\delta \in \Phi^\delta} \sum_{t=1}^T \left\langle \boldsymbol{u}^{(t)}, \phi^\delta(\boldsymbol{x}^{(t)}) - \boldsymbol{x}^{(t)} \right\rangle.$$

We are interested in algorithms whose regret is bounded by $\epsilon$ after $T = \text{poly}(N, 1/\epsilon)$ rounds. We refer to such algorithms as *fully polynomial no-regret learners*.

*Remark* 2.1. We clarify that all the algorithms we consider in this paper are *deterministic*, even when we allow mixed strategies. The fact that (1) contains an expectation over $\boldsymbol{x}^{(t)} \sim \pi^{(t)}$ is simply how $\Phi$-regret is defined; at no point does the algorithm actually sample from $\pi^{(t)}$. Using deterministic algorithms is in line with most of the prior work in the full feedback setting.

## 3 Overview of our results

In this section, we present an overview of our results on parameterized algorithms for minimizing $\Phi$-regret in extensive-form games. We shall first describe our results for the special case of Bayesian games with two actions per player, and we then treat general extensive-form games.

### 3.1 Bayesian games

For now, we assume that each player's strategy space is a hypercube $\{0, 1\}^N$. Hypercubes are linear transformations of tree-form decision problems; in particular, for Bayesian games in which each player has exactly two actions, the strategy space of every player is, up to linear transformations, a hypercube. Since our results are particularly clean for the hypercube case, we start with that.

First, we introduce the set of *depth-$k$ decision tree deviations* $\Phi_{\text{DT}}^k$, which can be described as follows. For each of $k \in \mathbb{N}$ rounds, the deviator first elects a decision point and receives a recommendation, whereupon the deviator gets to decide which action to follow in that decision point. More formally, the set of deviations $\Phi_{\text{DT}}^k$ is defined as follows:

1. The deviator observes an index $j_0 \in [N]$.
2. For $i = 1, \ldots, k$: the deviator selects an index $j_i \in [N]$, and observes $\boldsymbol{x}[j_i]$.
3. The deviator selects $a_0 \in \{0, 1\}$.

We call attention to the order of operations. In particular, each query $j$ is allowed to depend on previously observed $\boldsymbol{x}[j]$s. We can assume (WLOG) that the deviator always chooses $k$ distinct indices $j$. Now, the set of deviations $\phi : \{0, 1\}^N \to [0, 1]^N$ that can be expressed in the above manner is precisely the set of functions representable as (randomized) depth-$k$ decision trees on $N$ variables. To connect $\Phi_{\text{DT}}^k$ with the concepts referred to earlier, we clarify that $k = 1$ corresponds to linear-swap deviations, while $k = N$ captures all possible swap deviations. Our first result is a parameterized online algorithm minimizing regret with respect to deviations in $\Phi_{\text{DT}}^k$. (All our results are in the full feedback model under a strongly adaptive adversary.)

**Theorem 3.1.** *There is an online algorithm incurring (average) $\Phi_{\text{DT}}^k$-regret at most $\epsilon$ in $N^{O(k)}/\epsilon^2$ rounds with a per-round running time of $N^{O(k)}/\epsilon$.*

Next, we consider the set $\Phi^k_{\mathrm{poly}}$ consisting of all *degree-k* polynomials $\phi : \{0,1\}^N \to \{0,1\}^N$. Our result for this class of deviations mirrors the one for $\Phi^k_{\mathrm{DT}}$, but with a worse dependence on $k$.

**Theorem 3.2.** *There is an online algorithm incurring $\Phi^k_{\mathrm{poly}}$-regret at most $\epsilon$ in $N^{O(k^3)}/\epsilon^2$ rounds with a per-round running time of $N^{O(k^3)}/\epsilon$.*

We find those results surprising; we originally surmised that even for quadratic polynomials ($k = 2$) the underlying online problem would be hard in the regime where $\epsilon \leq \mathrm{poly}(1/N)$. We will elaborate on our technical approach for establishing those results in Section 4 coming up.

**Hardness in behavioral strategies**  A salient aspect of the previous results, which was intentionally blurred above, is that the learner is allowed to output a *mixed strategy*—a probability distribution over $\{0,1\}^N$. In stark contrast, and perhaps surprisingly, when the learner is constrained to output *behavioral* strategies, that is to say, points in $[0,1]^N$, we show that the problem immediately becomes PPAD-hard even for degree $k = 2$ (Theorem 3.3)—thereby being intractable under standard complexity assumptions. We are not aware of any such hardness results pertaining to a natural online learning problem, necessitating the use of mixed strategies.

The key connection behind our lower bound is an observation by Hazan and Kale [2007], which reveals that any $\Phi^\beta$-regret minimizer is inadvertedly able to compute approximate fixed points of any deviation in $\Phi^\beta$ (Proposition B.1). Computing fixed points is in general a well-known (presumably) intractable problem, being PPAD-hard. In our context, the set $\Phi^\beta$ does not contain arbitrary (Lipschitz continuous) functions $[0,1]^N \to [0,1]^N$, but instead contains multilinear functions from $[0,1]^N$ to $[0,1]^N$. To establish PPAD-hardness for our problem, we start with a *generalized circuit* (Definition I.3), and we show that all gates can be approximately simulated using exclusively gates involving multilinear operations (Proposition I.7); we defer the formal argument to Appendix I.1. As a result, we arrive at the following hardness result.

**Theorem 3.3.** *If a regret minimizer $\mathcal{R}$ outputs strategies in $[0,1]^N$, it is PPAD-hard to guarantee $\overline{\mathrm{Reg}}_{\Phi^\beta} \leq \epsilon/\sqrt{N}$, even with respect to low-degree deviations and an absolute constant $\epsilon > 0$.*

## 3.2   Extensive-form games

We next expand our scope to arbitrary extensive-form games. We will assume here that the branching factor $b$ of the game is 2—any game can be transformed as such by incurring a $\log b$ factor overhead in the depth $d$ of the game tree. Generalizing $\Phi^k_{\mathrm{DT}}$ described above, we introduce the set of $k$-*mediator deviations* $\Phi^k_{\mathrm{med}}$. Informally, the player here has access to $k$ distinct mediators, which the player can query at any time; a formal definition is given in Section 4. Once again, the case $k = 1$ corresponds to linear-swap deviations. Further, if $\mathcal{X}$ denotes the set of pure strategies, we let $\Phi^k_{\mathrm{poly}}$ denote the set of all degree-$k$ deviations $\mathcal{X} \to \mathcal{X}$. We establish similar parameterized results in extensive-form games, but which may now also depend on the depth of the game tree $d$.

**Theorem 3.4.** *There is an online algorithm incurring at most an $\epsilon$ $\Phi^k_{\mathrm{poly}}$ regret in $N^{O(kd)^3}/\epsilon^2$ rounds with a per-round running time of $N^{O(kd)^3}/\epsilon$. For $\Phi^k_{\mathrm{med}}$ both bounds instead scale as $N^{O(k)}$.*

We recall that $N$ here denotes the dimension of the strategy space. We further clarify that parameter $k$ appearing in $\Phi^k_{\mathrm{poly}}$ is different than the $k$ in $\Phi^k_{\mathrm{med}}$: the former refers to the degree of a polynomial, while the latter is the number of mediators. As all $k$-mediator deviations are degree-$k$ polynomials (but not vice versa), it is to be expected that the bound in the theorem above concerning the former is worse. For a fixed degree $k$ and assuming that the game tree is *balanced*, in the sense that $d = \mathrm{polylog}\, N$, Theorem 3.4 guarantees a quasipolynomial complexity with respect to $\Phi^k_{\mathrm{poly}}$, even when $\epsilon$ is itself inversely quasipolynomial. The complexity we obtain for $\Phi^k_{\mathrm{med}}$ is more favorable, being polynomial for any extensive-form game.[1] Finally, in light of the connection between no-regret learning and convergence to correlated equilibria, our results imply parameterized tractability of the equilibrium concepts induced by $\Phi^k_{\mathrm{med}}$ or $\Phi^k_{\mathrm{poly}}$ (see Appendix F.1 for a formal treatment).

---

[1]The bounds of Theorem 3.4 when $k \gg 1$—and in particular in the special case of swap regret—are inferior to the ones obtained by Dagan et al. [2024] and Peng and Rubinstein [2024]. As we explain in more detail later in Section 6, bridging those gaps is an interesting open problem.

# 4 Technical contributions

From a technical standpoint, our starting point is the familiar template of Gordon et al. [2008] for minimizing $\Phi$-regret, which consists of two key components. Accordingly, we split our technical overview into two parts.

## 4.1 Circumventing fixed points

The first key ingredient one requires in the framework of Gordon et al. [2008] is an algorithm for computing an approximate *fixed point* of any function within the set of deviations. In particular, if $\mathcal{X}$ is the set of pure strategies and $\operatorname{conv} \mathcal{X}$ is the convex hull of $\mathcal{X}$, we now work with functions $\Phi^\delta \ni \phi^\delta : \operatorname{conv} \mathcal{X} \to \operatorname{conv} \mathcal{X}$, so that fixed points exist by virtue of Brouwer's theorem.[2] As we discussed earlier, this fixed point computation is—at least in some sense—inherent: Hazan and Kale [2007] observed that minimizing $\Phi^\delta$-regret is computationally equivalent to computing approximate fixed points of transformations in $\Phi^\delta$. Specifically, an efficient algorithm minimizing $\Phi^\delta$-regret— with respect to any sequence of utilities—can be used to compute an approximate fixed point of any transformation in $\Phi^\delta$ (Proposition B.1 in Appendix B). Given that functions in $\Phi^\delta$ are generally nonlinear, this brings us to PPAD-hard territory (Theorem 3.3), seemingly contradicting the recent positive results of Dagan et al. [2024] and Peng and Rubinstein [2024].

As we have alluded to, it turns out that there is a delicate precondition on the reduction of Hazan and Kale [2007] that makes all the difference: computing approximate fixed points is only necessary if the learner outputs points on $\operatorname{conv} \mathcal{X}$. In stark contrast, a crucial observation that drives our approach is that a learner who selects a probability distribution over $\mathcal{X}$ does *not* have to compute (approximate) fixed points of functions in $\Phi$. Instead, we show that it is enough to determine what we refer to as an approximate fixed point *in expectation*. More precisely, for a deviation $\Phi \ni \phi : \mathcal{X} \to \operatorname{conv} \mathcal{X}$ with an efficient representation, it is enough to compute a distribution $\pi \in \Delta(\mathcal{X})$ such that $\mathbb{E}_{\boldsymbol{x} \sim \pi} \phi(\boldsymbol{x}) \approx \mathbb{E}_{\boldsymbol{x} \sim \pi} \boldsymbol{x}$. It is quite easy to compute an approximate fixed point in expectation: take any $\boldsymbol{x}_1 \in \operatorname{conv} \mathcal{X}$, and consider the sequence $\boldsymbol{x}_1, \ldots, \boldsymbol{x}_L \in \operatorname{conv} \mathcal{X}$ such that $\boldsymbol{x}_{\ell+1} := \mathbb{E}_{\boldsymbol{x}'_\ell \sim \delta(\boldsymbol{x}_\ell)} \phi(\boldsymbol{x}'_\ell)$ for all $\ell$, where $\delta : \operatorname{conv} \mathcal{X} \to \Delta(\mathcal{X})$ is a mapping such that $\mathbb{E}_{\boldsymbol{x}' \sim \delta(\boldsymbol{x})}[\boldsymbol{x}'] = \boldsymbol{x}$.[3] Then, for $\pi := \mathbb{E}_{\ell \in [L]}[\delta(\boldsymbol{x}_\ell)]$, we have

$$\mathbb{E}_{\boldsymbol{x} \sim \pi}[\phi(\boldsymbol{x}) - \boldsymbol{x}] = \frac{1}{L} \sum_{\ell=1}^{L} \mathbb{E}_{\boldsymbol{x}'_\ell \sim \delta(\boldsymbol{x}_\ell)}[\phi(\boldsymbol{x}'_\ell) - \boldsymbol{x}'_\ell] = \frac{1}{L} \mathbb{E}_{\boldsymbol{x}'_L \sim \delta(\boldsymbol{x}_L)}[\phi(\boldsymbol{x}'_L) - \boldsymbol{x}_1] = O\left(\frac{1}{L}\right).$$

This procedure can replace the fixed point oracle required by the template of Gordon et al. [2008], which is prohibitive when $\Phi$ contains nonlinear functions, as we formalize in Appendix C.

**Application to faster computation of correlated equilibria** In fact, even in normal-form games where considering linear deviations suffices, computing a fixed point is relatively expensive, amounting to solving a linear system, dominating the per-iteration complexity. Leveraging instead our new reduction, we obtain the fastest algorithm for computing an approximate correlated equilibrium in the moderate-precision regime (Corollary 4.1). In particular, let us focus for simplicity on $n$-player normal-form games with a succinct representation. Here, each player $i \in [n]$ selects as strategy a probability distribution $\pi_i \in \Delta(\mathcal{A}_i)$, where we recall that $\mathcal{A}_i$ is a finite set of available actions. The expected utility of player $i$ is given by $u_i(\pi_1, \ldots, \pi_n) := \mathbb{E}_{a_1 \sim \pi_1, \ldots, a_n \sim \pi_n}[u_i(a_1, \ldots, a_n)]$, where $u_i : \mathcal{A}_1 \times \cdots \times \mathcal{A}_n \to [-1, 1]$. We assume that there is an expectation oracle that computes the vector

$$(u_i(a_i, \pi_{-i}))_{i \in [n], a_i \in \mathcal{A}_i} \tag{2}$$

in time bounded by $\mathsf{EO}(n, A)$, where $A := \max_i |\mathcal{A}_i|$; it is known that $\mathsf{EO}(n, A) \leq \operatorname{poly}(n, A)$ for most interesting classes of succinct classes of games [Papadimitriou and Roughgarden, 2008]. Using our framework, we arrive at the following result.

---

[2] We recall that $\delta : \operatorname{conv} \mathcal{X} \to \Delta(\mathcal{X})$ is used to extend a map $\phi : \mathcal{X} \to \operatorname{conv} \mathcal{X}$ to a map $\phi^\delta : \operatorname{conv} \mathcal{X} \to \operatorname{conv} \mathcal{X}$.

[3] For technical reasons, it is more convenient to work with functions with domain $\mathcal{X}$, which is why we use a mapping $\delta$ to sample a point in $\mathcal{X}$ before applying $\phi$.

**Corollary 4.1.** *For any $n$-player game in normal form, there is an algorithm that computes an $\epsilon$-correlated equilibrium and runs in time*

$$O\left(\frac{A\log A}{\epsilon^2}\left(\mathsf{EO}(n, A) + n\frac{A^2}{\epsilon}\right)\right).$$

Assuming that the oracle call to (2) ($\mathsf{EO}(n, A)$) does not dominate the per-iteration running time—which is indeed the case in, for example, polymatrix games—Corollary 4.1 gives (to our knowledge) the fastest algorithm for computing $\epsilon$-correlated equilibria in the moderate-precision regime $1/A^{\frac{\omega}{2}-1} \leq \epsilon \leq 1/\log A$, where $\omega \approx 2.37$ is the exponent of matrix multiplication [Williams et al., 2024]; without fast matrix multiplication, which is widely impractical, the lower bound instead reads $\epsilon \geq 1/\sqrt{A}$. We provide a comparison with previous algorithms in Table 1 and defer the details to Appendix I.3. Finally, we stress that similar improvements can be obtained beyond normal-form games using our template; indeed, virtually all prior $\Phi$-regret minimizers rely on some fixed point operation.

Table 1: Time complexity for computing $\epsilon$-correlated equilibria in $n$-player normal-form games with $A$ actions per player. The second column suppresses absolute constants and polylogarithmic factors. For simplicity, issues related to bit complexity have been ignored (that is, we work in the RealRAM model of computation).

| Reference | Time complexity |
|---|---|
| Ours (Theorem C.7) | $\frac{A}{\epsilon^2}\left(\mathsf{EO}(n, A) + n\frac{A^2}{\epsilon}\right)$ |
| [Anagnostides et al., 2022, Daskalakis et al., 2021] | $\frac{A}{\epsilon}\left(\mathsf{EO}(n, A) + nA^\omega\right)$ |
| [Dagan et al., 2024, Peng and Rubinstein, 2024] | $nA\log^{1/\epsilon}(nA)$ |
| [Papadimitriou and Roughgarden, 2008] | $(nA)^c\mathsf{EO}(n, A)$ for $c \gg 1$ |
| [Huang and Pan, 2023] | $\frac{A^2}{\epsilon^2}(nA^\omega)$ |

Before moving on, it is worth stressing that the discrepancy that has arisen between operating over $\Delta(\mathcal{X})$ versus $\mathrm{conv}\,\mathcal{X}$ is quite singular when it comes to regret minimization in extensive-form games and beyond. Kuhn's theorem [Kuhn, 1953] is often invoked to argue about their equivalence, but in our setting it is the nonlinear nature of deviations in $\Phi$ that invalidates that equivalence.[4] To tie up the loose ends, we adapt the reduction of Hazan and Kale [2007] to show that minimizing $\Phi$-regret over $\Delta(\mathcal{X})$ necessitates computing approximate fixed points in expectation (Proposition C.3), and we observe that the reductions of Dagan et al. [2024] and Peng and Rubinstein [2024] are indeed compatible with computing approximate fixed points in expectation; the latter observation is made precise in Appendix F.3.

## 4.2 Regret minimization over the set of deviations $\Phi$

The second ingredient prescribed by Gordon et al. [2008] is an algorithm minimizing *external regret* but with respect to the *set of deviations* $\Phi$. The crux in this second step lies in the fact that, even in normal-form games, $\Phi$ contains at least an exponential number of deviations, so black-box reductions are of little use here. Instead, the problem boils down to appropriately leveraging the combinatorial structure of $\Phi$, as we explain below.

We will first describe our approach when $\mathcal{X} = \{0, 1\}^N$, and we then proceed with the more technical generalization to extensive-form games. The key observation here is that regret minimization over $\Phi_{\mathrm{DT}}^k$ can be viewed as a tree-form decision problem of size $N^{O(k)}$. Terminal nodes in this decision problem are identified by the original index $j_0 \in [N]$, the queries $j_1, \ldots, j_k \in [N]$, their replies $a_1, \ldots, a_k \in \{0, 1\}$, and finally the action $a_0 \in \{0, 1\}$ that is played. Each tree-form strategy $\boldsymbol{q}$ in this decision problem defines a function $\phi_{\boldsymbol{q}} : \mathcal{X} \to \mathrm{conv}\,\mathcal{X}$, which is computed by following the

---

[4]Kuhn's theorem is also invalidated in extensive-form games with *imperfect recall* [Piccione and Rubinstein, 1997, Tewolde et al., 2023, Lambert et al., 2019], in which there is also a genuine difference between mixed and behavioral strategies. In such settings, however, it is NP-hard to even minimize external regret.

strategy $\boldsymbol{q}$ through the decision problem. Formally, we have

$$\phi_{\boldsymbol{q}}(\boldsymbol{x})[j_0] = \sum_{j_1,a_1,\ldots,j_k,a_k} \boldsymbol{q}[j_0,j_1,a_1,\ldots,j_k,a_k,1] \prod_{i=1}^{k} \boldsymbol{x}[j_i,a_i]$$

where $\boldsymbol{x}[j_i,a_i] = \boldsymbol{x}[j_i]$ if $a_i = 1$, and $1 - \boldsymbol{x}[j_i]$ if $a_i = 0$. Hence $\phi_{\boldsymbol{q}}$ is a degree-$k$ polynomial in $\boldsymbol{x}$.

Now, since $\boldsymbol{q} \mapsto \phi_{\boldsymbol{q}}(\boldsymbol{x})[i]$ is linear, it follows that $\boldsymbol{q} \mapsto \langle \boldsymbol{u}, \phi_{\boldsymbol{q}}(\boldsymbol{x}) \rangle$ is also linear for any given $\boldsymbol{u} \in \mathbb{R}^n$. Therefore, a regret minimizer on $\Phi_{\mathrm{DT}}^k$ can be constructed starting from any regret minimizer for tree-form decision problems; for example, *counterfactual regret minimization* [Zinkevich et al., 2007], or any of its modern variants. This enables us to rely on usual techniques for dealing with such problems, eventually leading to a complexity bound of $N^{O(k)}$, as we formalize in Appendix D.

For the set of low-degree polynomials $\Phi_{\mathrm{poly}}^k$, we leverage a result from Boolean analysis relating (randomized) low-depth decision trees with low-degree polynomials, stated below.

**Theorem 4.2** (Midrijanis, 2004). *Every degree-$k$ polynomial $f : \{0,1\}^N \to \{0,1\}$ can be written as a decision tree of depth at most $2k^3$.*

In particular, this implies that $\Phi_{\mathrm{poly}}^k \subseteq \Phi_{\mathrm{DT}}^{2k^3}$. Consequently, low-degree polynomials can be reduced to low-depth decision trees, albeit with an overhead in the exponent.

Turning to general extensive-form games, we follow a similar blueprint, although there are now additional technical challenges. In particular, in what follows, to describe the set of deviations it will be convenient to introduce a new formalism related to tree-form decision problems.

**Definition 4.3.** The *dual* $\bar{\mathcal{X}}$ of $\mathcal{X}$ is the decision problem identical to $\mathcal{X}$, except that the decision points and observation points have been swapped.

**Definition 4.4.** The *interleaving* $\mathcal{X} \otimes \mathcal{Y}$ is the tree-form decision problem defined as follows. There is a state $\boldsymbol{s} = (s_1, s_2) \in \mathcal{S}_1 \times \mathcal{S}_2$. The root state is the tuple $(\varnothing, \varnothing)$. The decision problem is defined by the player being able to interact with *both* decision problems, in the following manner. At each state $\boldsymbol{s} = (s_1, s_2)$:

- If $s_1$ and $s_2$ are both terminal then so is $\boldsymbol{s}$. Otherwise:

- If either of the $s_i$s is an observation point, then so is $\boldsymbol{s}$. The children are the states $(s_i', s_{-i})$ where $s_i'$ is a child of $s_i$. (If both $s_i$s are observation points, both children $s_1', s_2'$ are selected simultaneously. This can only happen at the root.)

- Otherwise, $\boldsymbol{s}$ is a decision point. The player selects an index $i \in \{1, 2\}$ at which to act, and a child $s_i'$ to transition to. The next state is $(s_i', s_{-i})$.

In $\mathcal{X} \otimes \mathcal{Y}$, the same state $(s_1, s_2)$ can be reachable through possibly exponentially many paths, because the learner may choose to interleave actions in $\mathcal{X}$ with actions in $\mathcal{Y}$ in any order. Thus, each state $(s_1, s_2)$ corresponds to actually exponentially many histories in $\mathcal{X} \otimes \mathcal{Y}$. In the discussion below, we will therefore carefully distinguish between *histories* and *states*. In light of the above exponential gap between histories and states, it seems wasteful to represent $\mathcal{X} \otimes \mathcal{Y}$ as a tree. Indeed, Zhang et al. [2023] recently studied *DAG*-form decision problems, and showed that regret minimization on them is possible so long as the DAG obeys some natural properties.

Using the language we have now introduced, we can define the set of *k-mediator deviations* $\Phi_{\mathrm{med}}^k$ as the set of reduced strategies in the decision problem $\mathcal{X} \otimes \bar{\mathcal{X}}^{\otimes k}$. That is, the player has access to not one but $k$ mediators, all holding strategy $\boldsymbol{x}$, which the player can query at any time. This is a significant advantage over having just one mediator since the player can send different queries to each of the $k$ mediators (who must all reply according to $\boldsymbol{x}$), and therefore can learn more about the strategy $\boldsymbol{x}$ than it could have otherwise. We will call the responses given by the mediator *action recommendations*. For a graphical illustration of such deviations, we refer to Figure 2 (in Appendix E).

Reduced strategies $\boldsymbol{q} \in \pi(\mathcal{X} \otimes \bar{\mathcal{X}}^{\otimes k})$, once again, induce functions $\phi_{\boldsymbol{q}} : \mathcal{X} \to \mathrm{conv}\,\mathcal{X}$ given by

$$\phi_{\boldsymbol{q}}(\boldsymbol{x})[z] = \sum_{z_1,\ldots,z_k} \boldsymbol{q}[z,z_1,\ldots,z_k] \prod_{i=1}^{k} \boldsymbol{x}[z_i],$$

and in particular we have that $\phi_{\boldsymbol{q}}$ is a degree-$k$ polynomial. We define $\Phi_{\mathrm{med}}^k$ as the set of such deviations. For that set, we show that there is a reduction to a particular type of DAG-form decision problem of size $N^{O(k)}$. As we explained, that formulation is more suitable than tree-form decision problems when the number of possible histories far exceeds the number of states, which is precisely the case when the player is gradually querying multiple mediators as the game progresses.

Finally, we establish a reduction from low-degree polynomials to having few mediators; namely, we show that $\Phi_{\mathrm{poly}}^k \subseteq \Phi_{\mathrm{med}}^{O(kd)^3}$, where we recall that $d$ is the depth of the game tree. Our basic strategy is to again leverage the connection between low-depth decision trees and low-degree polynomials we described earlier (Theorem 4.2). To do so, we need to cast our problem in terms of functions $\{0,1\}^N \to \{0,1\}^N$ instead of $\mathcal{X} \to \mathcal{X}$. To that end, we first show how to *extend* a degree-$k$ function $f : \mathcal{X} \to \{0,1\}$ to a degree-$kd$ function $\bar{f} : \{0,1\}^N \to \{0,1\}$; that is, $\bar{f}$ coincides with $f$ on all points in $\mathcal{X} \subseteq \{0,1\}^N$ (Lemma E.7). This step is where the overhead factor $d$ comes from. The final technical piece is to show that if each component of $\phi : \mathcal{X} \to \mathcal{X}$ can be expressed using $K$ mediators, the same holds for $\phi$; the naive argument here incurs another factor of $d$, but we show that this is in fact not necessary. The details of the above argument are deferred to Appendix E.

## 5 Further related research

A key reference point is the result of Blum and Mansour [2007], and a generalization due to Gordon et al. [2008], which reduces minimizing swap regret to minimizing external regret. Specifically, for the probability simplex $\Delta(\mathcal{A})$, it maintains a separate external-regret minimizer, one for each action $a \in \mathcal{A}$. Both the per-iteration complexity and the number of iterations required is generally polynomial in $A := |\mathcal{A}|$. Therefore, in settings where $A$ is exponentially large in the natural parameters of the problem (such as extensive-form games) it does not appear that the reduction of Blum and Mansour [2007] is of much use. It is tempting to instead rely on the reduction of Stoltz and Lugosi [2005] for minimizing *internal* regret, a weaker notion than swap regret, which is nonetheless sufficient for (asymptotic) convergence to correlated equilibria. However, one should be careful when relying on internal regret in settings where $A$ is exponentially large; as we point out in Remark A.1, internal regret can be smaller than swap regret by up to a factor of $A$, so it is only meaningful when $\epsilon \leq 1/A$, a regime which is generally out of reach for regret minimization techniques when $A$ is exponentially large.

This gap motivated the new reduction by Dagan et al. [2024] and Peng and Rubinstein [2024], which we discussed earlier. Beyond extensive-form games, those reductions apply whenever it is possible to minimize external regret efficiently. The complexity of computing correlated equilibria beyond the regime where the precision parameter $\epsilon$ is an absolute constant remains a major open problem, generally conjectured to be hard [von Stengel and Forges, 2008]; the recent online lower bound in the adversarial setting [Daskalakis et al., 2024] provides further evidence in support of that conjecture.

As a result, most prior work has focused on more permissive equilibrium concepts, understood through the framework of $\Phi$-regret [Morrill et al., 2021a,b, Farina et al., 2022, Bai et al., 2022, Bernasconi et al., 2023, Noarov et al., 2023, Dudík and Gordon, 2009, Gordon et al., 2008, Fujii, 2023, Dann et al., 2023, Mansour et al., 2022, Sharma, 2024, Cai et al., 2024a]). In terms of the most recent developments, Farina and Pipis [2023] established efficient learning dynamics minimizing what is referred to as linear swap regret (*cf.* Dann et al. [2023], Fujii [2023] for related results in Bayesian games). The solution concept that arises from linear swap regret was later endowed with a natural mediator-based interpretation by Zhang et al. [2024], which can be viewed as a natural precursor to this work. Convergence to correlated equilibria has also attracted attention in the context of Markov (aka. stochastic) games (*e.g.*, [Cai et al., 2024b, Jin et al., 2021, Erez et al., 2023, Liu and Zhang, 2023], and references therein).

Moreover, as we explained earlier, our approach also gives rise to a faster algorithm for computing approximate correlated equilibria in a certain regime. As we discuss further in Appendix I.3, improving the per-iteration complexity of Blum and Mansour [2007] has received interest in prior work [Ito, 2020, Greenwald et al., 2006, Yang and Mohri, 2017] (see also [Huang and Pan, 2023, Huang et al., 2023]). The main bottleneck lies in the (approximate) computation of a stationary distribution of a Markov chain, which can be phrased as a linear system. It is worth noting that solving linear systems faster than matrix multiplication even for a crude approximation is precluded, at least

subject to fine-grained complexity assumptions [Bafna and Vyas, 2021]; we are not aware whether such hardness results are also known for computing the stationary distribution of a Markov chain.

Finally, although we have so far mostly directed our attention to the game-theoretic implication of minimizing swap (or indeed $\Phi$) regret, namely the celebrated connection with correlated equilibria in repeated games, the notion of swap regret is a fundamental solution concept in its own right more broadly in online learning and learning theory. Compared to the more common notion of external regret, swap regret gives rise to a more appealing notion of hindsight rationality; as such, it is often adopted as a behavioral assumption to model learning agents (*e.g.*, [Deng et al., 2019]). It is also fundamentally tied to the notion of *calibration* [Hu and Wu, 2024], and recently inspired work by Gopalan et al. [2023] in the context of multi-group fairness.

## 6 Conclusions and future research

We provided a new family of parameterized algorithms for minimizing $\Phi$-regret in extensive-form games. Our results capture perhaps the most natural class of functions interpolating between linear-swap and swap deviations, namely degree-$k$ deviations. Along the way, we refined the usual template for minimizing $\Phi$-regret—taught in many courses on algorithmic game theory and online learning—which revolves around (approximate) fixed points [Gordon et al., 2008, Blum and Mansour, 2007, Stoltz and Lugosi, 2005]. Instead, we showed that it suffices to rely on a relaxation that we refer to as an approximate fixed point in expectation, which—unlike actual fixed points—can always be computed efficiently. Our refinement of the usual template for minimizing $\Phi$-regret is of independent interest beyond extensive-form games. For example, it can speed up the computation of approximate correlated equilibria even in normal-form games, as it obviates the need to solve a linear system in every round. As in the recent works by Dagan et al. [2024] and Peng and Rubinstein [2024], a crucial feature of our approach is to allow the learner to select a distribution over pure strategies, for otherwise we showed that regret minimization immediately becomes PPAD-hard (under a strongly adaptive adversary).

There are many interesting avenues for future research. First, the complexity of our algorithm pertaining to degree-$k$ deviations depends exponentially on the depth of the game tree. We suspect that such a dependency could be superfluous. To show this, it would be enough to refine Lemma E.7 by coming up with an extension whose degree does not depend on the depth of the game tree. It would also be interesting to devise parameterized algorithms for $k$-mediator deviations that recover as a special case the PTAS of Peng and Rubinstein [2024] and Dagan et al. [2024], so as to smoothly interpolate between existing results for linear-swap regret [Farina and Pipis, 2023] and the aforementioned results for swap regret; is $k = \tilde{O}(1/\epsilon)$ enough to capture swap regret?

Finally, perhaps the most important question is to understand the computational complexity of computing $\Phi$-equilibria in extensive-form games. In particular, our results raise the interesting question of whether there is an algorithm (in the centralized model) for computing in polynomial time an *exact* correlated equilibrium induced by low-degree deviations. Extending the paradigm of Papadimitriou and Roughgarden [2008] in that setting presents several challenges, not least because computing fixed points—which are crucial for implementing the separation oracle [Papadimitriou and Roughgarden, 2008]—is now computationally hard. Relatedly, we suspect that there is an inherent connection between fixed points and correlated equilibria, in the spirit of the equivalence between $\Phi$-regret minimization and fixed points established by Hazan and Kale [2007].

## Acknowledgements

We are grateful to the anonymous reviewers at NeurIPS for their helpful feedback. This material is based on work supported by the Vannevar Bush Faculty Fellowship ONR N00014-23-1-2876, National Science Foundation grants RI-2312342 and RI-1901403, ARO award W911NF2210266, and NIH award A240108S001.

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

# A    Further preliminaries

In this section, we introduce some further preliminaries. For additional background, we refer the interested reader to the excellent books of Cesa-Bianchi and Lugosi [2006] and Shoham and Leyton-Brown [2009]. Before we describe more formally the construction of Gordon et al. [2008], we make a remark regarding minimizing internal regret in extensive-form games.

*Remark* A.1 (Swap versus internal regret). When it comes to defining correlated equilibria in normal-form games, there are two prevalent definitions appearing in the literature; one is based on *internal regret*, while the other on *swap regret* (*e.g.*, [Ganor and Karthik C. S., 2018, Goldberg and Roth, 2016]). The key difference is that internal regret only contains deviations that swap a *single action*—thereby being weaker. Nevertheless, it is not hard to see that swap regret can only be larger by a factor of $|\mathcal{X}|$ [Blum and Mansour, 2007], where we recall that $\mathcal{X}$ denotes the set of pure strategies. So, in normal-form games those two definitions are polynomially equivalent, and in most applications one can safely switch from one to the other.

However, this is certainly not the case in games with an exponentially large action space, such as extensive-form games. In fact, the definition of internal regret itself is problematic when the action set is exponentially large: the uniform distribution always attains an error of at most $1/|\mathcal{X}|$. Consequently, any guarantee for $\epsilon \geq 1/|\mathcal{X}|$ is vacuous. That is, if $|\mathcal{X}|$ is exponentially large, an algorithm that requires a number of iterations polynomial in $1/\epsilon$—which is what we expect to get from typical no-regret dynamics—would need an exponential number of iterations to yield a non-trivial guarantee; this issue with internal regret was also observed by Fujii [2023]. Nevertheless, internal regret in the context of games with an exponentially large action set was used in a recent work by Chen et al. [2023], who provided oracle-efficient algorithms for minimizing internal regret.

## A.1    The construction of Gordon et al. [2008]

Gordon et al. [2008], building on earlier work by Blum and Mansour [2007] and Stoltz and Lugosi [2005], came up with a general recipe for minimizing $\Phi^\delta$-regret. That construction relies on a no-regret learning algorithm on the set of deviations $\Phi^\delta$, which we denote by $\mathcal{R}_\Phi$. Then, a $\Phi^\delta$-regret minimizer on $\mathrm{conv}\,\mathcal{X}$ can be constructed as follows: on each iteration $t = 1, \ldots, T$, the learner performs the following steps.

1. Receive $\phi^{(t)}$ from $\mathcal{R}_\Phi$. Select $\boldsymbol{x}^{(t)} \in \mathrm{conv}\,\mathcal{X}$ as an $\epsilon$-fixed point of $\phi^{(t)}$: $\|\phi^{(t)}(\boldsymbol{x}^{(t)}) - \boldsymbol{x}^{(t)}\|_{\mathcal{X}} \leq \epsilon$.

2. Upon receiving utility $\boldsymbol{u}^{(t)} \in \mathcal{U}$, pass utility $\Phi^\delta \ni \phi^\delta \mapsto \langle \boldsymbol{u}^{(t)}, \phi^\delta(\boldsymbol{x}^{(t)})\rangle$ to $\mathcal{R}_\Phi$.

The main guarantee regarding the above algorithm is summarized below.

**Theorem A.2** (Gordon et al., 2008). *Suppose that* $\overline{\mathrm{Reg}}^T$ *is the external regret incurred by* $\mathcal{R}_\Phi$. *After* $T$ *rounds of the above algorithm, we have*

$$\max_{\phi^\delta \in \Phi^\delta} \frac{1}{T} \sum_{t=1}^{T} \left\langle \boldsymbol{u}^{(t)}, \phi^\delta(\boldsymbol{x}^{(t)}) - \boldsymbol{x}^{(t)} \right\rangle \leq \overline{\mathrm{Reg}}^T + \epsilon.$$

In Appendix C.1, we will relax the requirement of needing (approximate) fixed points, while at the same time maintaining the guarantee of Theorem A.2.

# B    Hardness of minimizing $\Phi$-regret in behavioral strategies

In this section, we show that if the learner is constrained to output in reach round a strategy in $\mathrm{conv}\,\mathcal{X}$, then there is no efficient algorithm (under standard complexity assumptions) minimizing $\Phi^\beta$-regret (Theorem 3.3); here, $\beta : \mathrm{conv}\,\mathcal{X} \to \Delta(\mathcal{X})$ is the behavioral strategy mapping (introduced in the sequel as Definition C.5), the expression of which is not important for the purpose of this section. The key connection is a result by Hazan and Kale [2007], showing that any $\Phi^\beta$-regret minimizer is able to compute approximate fixed points of any deviations in $\Phi^\beta$. We then show that the set of induced deviations, even on the hypercube $\mathcal{X} = \{0, 1\}^N$, is rich enough to approximate PPAD-hard fixed-point problems.

In this context, consider a transformation $\Phi^\beta \ni \phi^\beta : [0,1]^N \to [0,1]^N$ for which we want to compute an approximate fixed point $\boldsymbol{x} \in \text{conv}\,\mathcal{X}$; that is, $\|\phi^\beta(\boldsymbol{x}) - \boldsymbol{x}\|_2 \leq \epsilon$, for some precision parameter $\epsilon > 0$. (It is convenient in the construction below to measure the fixed-point error with respect to $\|\cdot\|_2$.) Hazan and Kale [2007] observed that a $\Phi^\beta$-regret minimizer can be readily turned into an algorithm for computing fixed points of any function in $\Phi^\beta$, as stated formally below. Before we proceed, we remind that here and throughout we operate under a strongly adaptive adversary, which is quite crucial in the construction of Hazan and Kale [2007].

**Proposition B.1** (Hazan and Kale, 2007). *Consider a regret minimizer $\mathcal{R}$ operating over $[0,1]^N$. If $\mathcal{R}$ runs in time $\text{poly}(N, 1/\epsilon)$ and guarantees $\overline{\text{Reg}}_{\Phi^\beta}^T \leq \epsilon$ for any sequence of utilities, then there is a $\text{poly}(N, 1/\epsilon)$ algorithm for computing an $(\epsilon\sqrt{N})$-fixed point of any $\phi^\beta \in \Phi^\beta$ with respect to $\|\cdot\|_2$, assuming that $\phi^\beta$ can be evaluated in polynomial time.*

Proposition B.1 significantly circumscribes the class of problems for which efficient $\Phi^\beta$-regret minimization is possible, at least when operating in behavioral strategies. Indeed, computing fixed points is in general a well-known (presumably) intractable problem. In our context, the set $\Phi^\beta$ does not contain arbitrary (Lipschitz continuous) functions $[0,1]^N \to [0,1]^N$, but instead contains multilinear functions from $[0,1]^N$ to $[0,1]^N$. Nonetheless, we show that PPAD-hardness persists in our setting. The basic idea is as follows. We start with a *generalized circuit* (Definition I.3), and we show that all gates can be approximately simulated using exclusively gates involving multilinear operations (Proposition I.7). The proof of that claim appears in Appendix I.1. As a result, we arrive at the main hardness result of this section, restated below.

**Theorem 3.3.** *If a regret minimizer $\mathcal{R}$ outputs strategies in $[0,1]^N$, it is PPAD-hard to guarantee $\overline{\text{Reg}}_{\Phi^\beta} \leq \epsilon/\sqrt{N}$, even with respect to low-degree deviations and an absolute constant $\epsilon > 0$.*

We also obtain a stronger hardness result under a stronger complexity assumption put forward by Babichenko et al. [2016] (Theorem I.9). At first glance, it may seem that the above results are at odds with the recent positive results of Dagan et al. [2024] and Peng and Rubinstein [2024], which seemingly obviate the need to compute approximate fixed points. As we have alluded to, the key restriction that drives Theorem 3.3 lies in constraining the learner to output behavioral strategies. In the coming section, we show that there is an interesting twist which justifies the discrepancy highlighted above.

## C  Circumventing fixed points

The previous section, and in particular Theorem 3.3, seems to preclude the ability to minimize $\Phi$-regret efficiently when the set of (extended) deviations contains nonlinear functions.[5] In this section, we will show how to circumvent this issue via a relaxed notion of what constitutes a fixed point (Definition C.1). In the sequel, we will work with deviations $\phi$ with domain $\mathcal{X}$ instead of $\text{conv}\,\mathcal{X}$.

### C.1  Approximate expected fixed points

The key to our construction is to allow the learner to play *distributions* over $\mathcal{X}$, not merely points in $\text{conv}\,\mathcal{X}$, and to use a relaxed notion of a fixed point, formally introduced below.

**Definition C.1.** We say that a distribution $\pi \in \Delta(\mathcal{X})$ is an $\epsilon$-*expected fixed point* of $\phi \in (\text{conv}\,\mathcal{X})^{\mathcal{X}}$ if $\|\mathbb{E}_{\boldsymbol{x}\sim\pi}[\phi(\boldsymbol{x}) - \boldsymbol{x}]\|_{\mathcal{X}} \leq \epsilon$.

The key now is to replace the fixed point oracle in the framework of Gordon et al. [2008] (recalled in Appendix A) with an oracle that instead returns an $\epsilon$-fixed point in expectation per Definition C.1. The learner otherwise proceeds as in the algorithm of Gordon et al. [2008] (our overall construction is spelled out as Algorithm 1 in Appendix I.2). It is easy to show, following the proof of Gordon et al. [2008], that a fixed point in expectation is still sufficient to minimize $\Phi$-regret.

**Theorem C.2** ($\Phi$-regret with $\epsilon$-expected fixed points). *Suppose that the external regret of $\mathcal{R}_\Phi$ over $\Phi$ after $T$ repetitions is at most $\overline{\text{Reg}}^T$. Then, the $\Phi$-regret of Algorithm 1 can be bounded as $\overline{\text{Reg}}^T + \epsilon$.*

Analogously to Proposition B.1, it turns out that there is a certain equivalence between minimizing $\Phi$ in $\Delta(\mathcal{X})$ and computing *expected* fixed points:

---

[5]For linear functions, fixed points can be computed exactly via a linear program.

**Proposition C.3.** *Consider a regret minimizer $\mathcal{R}$ operating over $\Delta(\mathcal{X})$. If $\mathcal{R}$ runs in time* $\mathsf{poly}(N, 1/\epsilon)$ *and guarantees* $\overline{\mathrm{Reg}}_\Phi^T \leq \epsilon$ *for any sequence of utilities, then there is a* $\mathsf{poly}(N, 1/\epsilon)$ *algorithm for computing* $(\epsilon D_\mathcal{X})$*-expected fixed points of* $\phi \in \Phi$*, assuming that we can efficiently compute* $\mathbb{E}_{\boldsymbol{x}^{(t)} \sim \pi^{(t)}}[\phi(\boldsymbol{x}^{(t)}) - \boldsymbol{x}^{(t)}]$ *at any time t. Here, $D_\mathcal{X}$ is the diameter of $\mathcal{X}$ with respect to* $\| \cdot \|_2$.

The proof proceeds similarly to Proposition B.1, and so we include it in Appendix I.2. Next, we present a method for computing approximate expected fixed points of functions $\phi \in \Phi$ without having to solve a PPAD-hard problem.

## C.2 Extending deviation maps to $\mathrm{conv}\,\mathcal{X}$

First, since we will work both over $\mathrm{conv}\,\mathcal{X}$ and distributions in $\Delta(\mathcal{X})$, we need efficient methods for passing between them. To that end, we introduce the following notion.

**Definition C.4.** A map $\delta : \mathrm{conv}\,\mathcal{X} \to \Delta(\mathcal{X})$ is

- *consistent* if $\mathbb{E}_{\boldsymbol{x}' \sim \delta(\boldsymbol{x})}\, \boldsymbol{x}' = \boldsymbol{x}$, and

- *efficient* if, given some $\phi \in \Phi$ and $\boldsymbol{x} \in \mathrm{conv}\,\mathcal{X}$, it is easy to compute $\phi^\delta(\boldsymbol{x}) := \mathbb{E}_{\boldsymbol{x}' \sim \delta(\boldsymbol{x})}\, \phi(\boldsymbol{x}')$.

We will call the map $\phi^\delta : \mathrm{conv}\,\mathcal{X} \to \mathrm{conv}\,\mathcal{X}$ the *extended map* of $\phi$.

One may ask why we use this indirect method of defining $\phi^\delta$ rather than simply directly using the representation of $\phi$ (for example, as a polynomial) to extend $\phi$ to $\mathrm{conv}\,\mathcal{X}$. The answer is that, even assuming that $\phi : \mathcal{X} \to \mathcal{X}$ is represented as a multilinear polynomial (which is the representation assumed in the majority of this paper), naively extending that polynomial to domain $\mathrm{conv}\,\mathcal{X}$ will not necessarily result in a function $\bar{\phi} : \mathrm{conv}\,\mathcal{X} \to \mathrm{conv}\,\mathcal{X}$. For an example, consider the decision problem $\mathcal{X}$ depicted in Figure 1, and consider the function $\phi : \mathcal{X} \to \mathcal{X}$ given by $\phi(\boldsymbol{x}) = (x_1 + x_3, x_2x_4, x_2x_5, x_2, 0)$. One can easily check by hand that $\phi$ is indeed a function $\mathcal{X} \to \mathcal{X}$, but also that, for the strategy $\boldsymbol{x} = (1/2, 1/2, 0, 1/2, 0) \in \mathrm{conv}\,\mathcal{X}$, we have $\phi(\boldsymbol{x}) = (1/2, 1/4, 0, 1/2, 0) \notin \mathrm{conv}\,\mathcal{X}$. Thus, we need a more robust way of extending functions $\mathcal{X} \to \mathrm{conv}\,\mathcal{X}$ to functions $\mathrm{conv}\,\mathcal{X} \to \mathrm{conv}\,\mathcal{X}$, ideally one that is dependent only the function $\phi$, not its representation.

We now give two methods of constructing consistent and efficient maps $\delta : \mathrm{conv}\,\mathcal{X} \to \Delta(\mathcal{X})$ for tree-form strategy sets $\mathcal{X}$. The first is the behavioral strategy map.

**Definition C.5.** The *behavioral strategy map* $\beta : \mathrm{conv}\,\mathcal{X} \to \Delta(\mathcal{X})$ is defined as follows: $\beta(\boldsymbol{x})$ is the distribution of pure strategies generated by sampling, at each decision point $j$ for which $\boldsymbol{x}[j] > 0$, an action $a$ according to the probabilities $\boldsymbol{x}[ja]/\boldsymbol{x}[j]$. Formally,

$$\beta(\boldsymbol{x})[\boldsymbol{y}] := \prod_{ja:\boldsymbol{x}[j]>0,\boldsymbol{y}[ja]=1} \frac{\boldsymbol{x}[ja]}{\boldsymbol{x}[j]}.$$

It is possible for $\phi^\beta$ to be not a polynomial even when $\phi$ is a polynomial, because $\beta$ is *itself* not a polynomial. It is clear that $\beta$ is consistent. For efficiency, we show the following claim.

**Proposition C.6.** *Let $\beta : \mathrm{conv}\,\mathcal{X} \to \Delta(\mathcal{X})$ be the behavioral strategy map. Let $\phi : \mathcal{X} \to \mathrm{conv}\,\mathcal{X}$ be expressed as a polynomial of degree at most $k$, in particular, as a sum of at most $O(N^k)$ terms. Then there is an algorithm running in time $N^{O(k)}$ that, given $\phi$ and $\boldsymbol{x} \in \mathrm{conv}\,\mathcal{X}$, computes $\phi^\beta(\boldsymbol{x})$.*

*Proof.* To compute $\mathbb{E}_{\boldsymbol{x}' \sim \beta(\boldsymbol{x})}\, \phi(\boldsymbol{x}')$, since $\phi$ is a polynomial, it suffices to compute $\mathbb{E}_{\boldsymbol{x}' \sim \beta(\boldsymbol{x})}\, m(\boldsymbol{x}')$ for multilinear monomials $m$ of degree at most $K$, that is, functions of the form $m_S(\boldsymbol{x}) := \prod_{z \in S} \boldsymbol{x}[z]$ where $S \subseteq \mathcal{Z}$ has size at most $k$. There are two cases. First, there are monomials that are clearly identically zero: in particular, if there are two nodes $ja, ja' \preceq S$ for $a \neq a'$, then $m_S \equiv 0$ because a player cannot play two different actions at $j$. For monomials that are not identically zero, we have

$$\mathbb{E}_{\boldsymbol{x}' \sim \beta(\boldsymbol{x})} \prod_{ja \in S} \boldsymbol{x}[ja] = \prod_{ja \preceq S:\boldsymbol{x}[j]>0} \frac{\boldsymbol{x}[ja]}{\boldsymbol{x}[j]},$$

which is computable in time $O(kd)$. Thus, the overall time complexity is $O(kdN^k) \leq N^{O(k)}$. $\qquad\square$

The behavioral strategy map is in some sense the *canonical* strategy map: when one writes a tree-form strategy $\boldsymbol{x} \in \operatorname{conv} \mathcal{X}$ without further elaboration on what distribution $\Delta(\mathcal{X})$ it is meant to represent, it is often implicitly or explicitly assumed to mean the behavioral strategy.

The behavioral strategy map has the unfortunate property that it usually outputs distributions of exponentially-large support; indeed, if $\boldsymbol{x} \in \operatorname{relint} \operatorname{conv} \mathcal{X}$ then $\beta(\boldsymbol{x})$ is *full*-support.

The second example we propose, which we call a *Carathéodory map*, always outputs low-support distributions. In particular, for any $\boldsymbol{x} \in \operatorname{conv} \mathcal{X}$, Carathéodory's theorem on convex hulls guarantees that $\boldsymbol{x}$ is a convex combination of $N$ pure strategies[6] $\boldsymbol{x}_1, \ldots, \boldsymbol{x}_N \in \mathcal{X}$. Grötschel et al. [1981, Theorem 3.9] moreover showed that there exists an efficient algorithm for computing the appropriate convex combination. Thus, fixing some efficient algorithm for this computational problem, we define a *Carathéodory map* $\gamma : \operatorname{conv} \mathcal{X} \to \Delta(\mathcal{X})$ to be any consistent map that returns a distribution of support at most $N$. Given such a mapping, computing $\phi^\gamma(\boldsymbol{x})$ is easy: one simply writes $\boldsymbol{x} = \sum_i \alpha_i \boldsymbol{x}_i$ by computing $\gamma(\boldsymbol{x})$, and returns $\phi^\gamma(\boldsymbol{x}) = \sum_i \alpha_i \phi(\boldsymbol{x}_i)$. This only requires a $\operatorname{poly}(N)$-time computation of $\gamma$, and $N$ evaluations of the function $\phi$. As before, when $\phi$ is a degree-$k$ polynomial, the time complexity of computing $\phi^\gamma$ is bounded by $N^{O(k)}$.

### C.3 Efficiently computing fixed points in expectation

Now let $\delta : \operatorname{conv} \mathcal{X} \to \Delta(\mathcal{X})$ be consistent and efficient. Consider the following algorithm. Given $\phi \in \Phi$, select $\boldsymbol{x}_1 \in \operatorname{conv} \mathcal{X}$ arbitrarily, and then for each $\ell > 1$ set $\boldsymbol{x}_\ell := \phi^\delta(\boldsymbol{x}_{\ell-1})$. Finally, select $\pi := \mathbb{E}_{\ell \sim [L]} \delta(\boldsymbol{x}_\ell) \in \Delta(\mathcal{X})$ as the output distribution. By a telescopic cancellation, we have

$$\left\| \mathop{\mathbb{E}}_{\boldsymbol{x} \sim \pi} [\phi(\boldsymbol{x}) - \boldsymbol{x}] \right\|_{\mathcal{X}} = \frac{1}{L} \left\| \sum_{\ell=1}^{L} \mathop{\mathbb{E}}_{\boldsymbol{x} \sim \delta(\boldsymbol{x}_\ell)} [\phi(\boldsymbol{x}) - \boldsymbol{x}] \right\|_{\mathcal{X}} \leq \frac{1}{L} \left\| \mathop{\mathbb{E}}_{\boldsymbol{x} \sim \delta(\boldsymbol{x}_L)} [\phi(\boldsymbol{x}) - \boldsymbol{x}_1] \right\|_{\mathcal{X}} \leq \frac{2}{L},$$

as desired. As a result, applying Theorem C.2, we arrive at the following conclusion.

**Theorem C.7.** *Let $\mathcal{R}_\Phi$ be an regret minimizer on $\Phi$ whose external regret after $T$ iterations is $\overline{\operatorname{Reg}}^T$ and whose per-iteration runtime is $R_1$, and assume that evaluating the extended map $\phi^\delta :$ $\operatorname{conv} \mathcal{X} \to \operatorname{conv} \mathcal{X}$ takes time $R_2$. Then, for every $\epsilon > 0$, there is a learning algorithm on $\mathcal{X}$ whose $\Phi$-regret after $T$ iterations is at most $\overline{\operatorname{Reg}}^T + \epsilon$ and whose per-iteration runtime is $O(R_1 + R_2/\epsilon)$.*

The above result provides a full black-box reduction from $\Phi$-regret minimization to external regret minimization on $\Phi$, with no need for the possibly-expensive computation of a fixed point. We note that the iterates of the algorithm will depend on the choice of $\delta$—for example, setting $\delta = \beta$ and setting $\delta = \gamma$ will produce different iterates.

## D Low-degree regret on the hypercube

In this section, we let $\mathcal{X}$ be the hypercube $\{0, 1\}^N$. For the convenience of the reader, we first recall that the set of deviations $\Phi_{\mathrm{DT}}^k$ is defined as follows:

1. The deviator observes an index $j_0 \in [N]$.
2. For $i = 1, \ldots, k$: The deviator selects an index $j_i \in [N]$, and observes $\boldsymbol{x}[j_i]$.
3. The deviator selects $a_0 \in \{0, 1\}$.

As we observed earlier, the above process describes a tree-form decision problem of size $N^{O(k)}$. In particular, terminal nodes in this decision problem are identified by the original index $j_0 \in [N]$, the queries $j_1, \ldots, j_k \in [N]$, their replies $a_1, \ldots, a_k \in \{0, 1\}$, and finally the action $a_0 \in \{0, 1\}$ that is played. Each tree-form strategy $\boldsymbol{q}$ in this decision problem defines a function $\phi_{\boldsymbol{q}} : \mathcal{X} \to \operatorname{conv} \mathcal{X}$, which is computed by following the strategy $\boldsymbol{q}$ through the decision problem. Namely,

$$\phi_{\boldsymbol{q}}(\boldsymbol{x})[j_0] = \sum_{j_1, a_1, \ldots, j_k, a_k} \boldsymbol{q}[j_0, j_1, a_1, \ldots, j_k, a_k, 1] \prod_{i=1}^{k} \boldsymbol{x}[j_i, a_i]$$

---

[6]Applying Carathéodory naively would give $N + 1$ instead of $N$, but we can save 1 because the tree-form strategy set is never full-dimensional as a subset of $\{0, 1\}^N$.

where $\boldsymbol{x}[j_i, a_i] = \boldsymbol{x}[j_i]$ if $a_i = 1$, and $1 - \boldsymbol{x}[j_i]$ if $a_i = 0$. We see that $\phi_{\boldsymbol{q}}$ is a degree-$k$ polynomial in $\boldsymbol{x}$.

We define $\Phi_{\mathrm{DT}}^k$ as the set of such functions $\phi_{\boldsymbol{q}}$. The "DT" in the name $\Phi_{\mathrm{DT}}^k$ stands for *decision tree*: the set of functions $\phi : \mathcal{X} \to \mathrm{conv}\,\mathcal{X}$ that can be expressed in the above manner is precisely the set of functions representable as (randomized) depth-$k$ decision trees on $N$ variables.

For intuition, we mention the following special cases:

- $\Phi_{\mathrm{DT}}^0$ is the set of external deviations.
- $\Phi_{\mathrm{DT}}^1$ is the set of all single-query deviations, which Fujii [2023] showed to be equivalent to the set of all linear deviations when $\mathcal{X}$ is a hypercube.
- $\Phi_{\mathrm{DT}}^N$ is the set of all swap deviations.

Since $\boldsymbol{q} \mapsto \phi_{\boldsymbol{q}}(\boldsymbol{x})[i]$ is linear, it follows that $\boldsymbol{q} \mapsto \langle \boldsymbol{u}, \phi_{\boldsymbol{q}}(\boldsymbol{x}) \rangle$ is also linear for any given $\boldsymbol{u} \in \mathbb{R}^n$. Therefore, a regret minimizer on $\Phi_{\mathrm{DT}}^k$ can be constructed starting from any regret minimizer for tree-form decision problems, such as counterfactual regret minimization [Zinkevich et al., 2007].

**Proposition D.1.** *There is a $N^{O(k)}$-time-per-round regret minimizer on $\Phi_{\mathrm{DT}}^k$ whose external regret is at most $\epsilon$ after $N^{O(k)}/\epsilon^2$ rounds.*

Thus, combining with Proposition C.6 and Theorem C.7, we immediately obtain a $\Phi_{\mathrm{DT}}^k$-regret minimizer with the following complexity.

**Corollary D.2.** *There is a $N^{O(k)}/\epsilon$-time-per-round regret minimizer on $\mathcal{X}$ whose $\Phi_{\mathrm{DT}}^k$-regret is at most $\epsilon$ after $N^{O(k)}/\epsilon^2$ rounds.*

Next, we relate depth-$k$ decision trees to low-degree polynomials. Let $\Phi_{\mathrm{poly}}^k$ be the set of degree-$k$ polynomials $\phi : \mathcal{X} \to \mathcal{X}$. We appeal to a result from the literature on Boolean analysis, recalled below.

**Theorem 4.2** (Midrijanis, 2004)**.** *Every degree-$k$ polynomial $f : \{0,1\}^N \to \{0,1\}$ can be written as a decision tree of depth at most $2k^3$.*

In particular, $\Phi_{\mathrm{poly}}^k \subseteq \Phi_{\mathrm{DT}}^{2k^3}$. Corollary D.2 thus also implies a $\Phi_{\mathrm{poly}}^k$-regret minimizer:

**Corollary D.3.** *Let $\mathcal{X} = \{0,1\}^N$. There is an $N^{O(k^3)}/\epsilon$-time-per-round regret minimizer on $\mathcal{X}$ whose $\Phi_{\mathrm{poly}}^k$-regret is at most $\epsilon$ after $N^{O(k^3)}/\epsilon^2$ rounds.*

It is reasonable to ask whether the above result generalizes to polynomials $\phi : \mathcal{X} \to \mathrm{conv}\,\mathcal{X}$. Indeed, when $k \le 1$ or $k = N$, every degree-$k$ polynomials $\phi : \mathcal{X} \to \mathrm{conv}\,\mathcal{X}$ can be written as a convex combination of degree-$k$ polynomials $\phi : \mathcal{X} \to \mathcal{X}$, even for arbitrary tree-form decision problems.[7] However, this is not generally true. A brute-force search shows that the polynomial $\phi : \{0,1\}^4 \to [0,1]^4$ given by

$$\phi(x_1, x_2, x_3, x_4) = x_1 - x_1 x_2 - \frac{1}{2} x_1 x_3 + \frac{1}{2} x_2 x_3 + \frac{1}{2} x_3 x_4$$

is quadratic, but it is not a convex combination of quadratics whose range is $\{0,1\}^4$. Perhaps more glaringly, if one could efficiently represent the set of quadratic functions $\phi : \{0,1\}^N \to [0,1]^N$, then one could in particular *decide* whether a given quadratic function $\phi : \{0,1\}^N \to \mathbb{R}^N$ has range $[0,1]^N$. But this is a coNP-complete problem.

## E    Extensive-form games

The goal of this section is to extend the results in the previous section to the *extensive-form* setting, that is, to generalize them to all tree-form decision problems.

---

[7]For degree 0 and $N$ this is trivial; for degree 1 it is due to Zhang et al. [2024].

### E.1 Interleaving decision problems

We first recall the operations of merging decision problems that will be very useful as notation in the subsequent discussion. In particular, given two decision problems $\mathcal{X}$ and $\mathcal{Y}$ with node sets $\mathcal{S}_1$ and $\mathcal{S}_2$ respectively, we restate the following definitions previously introduced in Section 4.

**Definition 4.3.** The *dual* $\bar{\mathcal{X}}$ of $\mathcal{X}$ is the decision problem identical to $\mathcal{X}$, except that the decision points and observation points have been swapped.

**Definition 4.4.** The *interleaving* $\mathcal{X} \otimes \mathcal{Y}$ is the tree-form decision problem defined as follows. There is a state $\boldsymbol{s} = (s_1, s_2) \in \mathcal{S}_1 \times \mathcal{S}_2$. The root state is the tuple $(\varnothing, \varnothing)$. The decision problem is defined by the player being able to interact with *both* decision problems, in the following manner. At each state $\boldsymbol{s} = (s_1, s_2)$:

- If $s_1$ and $s_2$ are both terminal then so is $\boldsymbol{s}$. Otherwise:

- If either of the $s_i$s is an observation point, then so is $\boldsymbol{s}$. The children are the states $(s'_i, s_{-i})$ where $s'_i$ is a child of $s_i$. (If both $s_i$s are observation points, both children $s'_1, s'_2$ are selected simultaneously. This can only happen at the root.)

- Otherwise, $\boldsymbol{s}$ is a decision point. The player selects an index $i \in \{1, 2\}$ at which to act, and a child $s'_i$ to transition to. The next state is $(s'_i, s_{-i})$.

It follows immediately from definitions that $\bar{\bar{\mathcal{X}}} = \mathcal{X}$, and $\otimes$ is associative and commutative. The name and notation for the dual is inspired by the observation that $\langle \boldsymbol{x}, \boldsymbol{y} \rangle = 1$ for all $\boldsymbol{x} \in \mathcal{X}$ and $\boldsymbol{y} \in \bar{\mathcal{X}}$: indeed, the component-wise product $\boldsymbol{x}[z]\boldsymbol{y}[z]$ is exactly the probability that one reaches terminal node $z$ by following strategy $\boldsymbol{x}$ at $\mathcal{X}$'s decision points and $\boldsymbol{y}$ at $\mathcal{X}$'s observation points. We also define the notation $\mathcal{X}^{\otimes k} := \mathcal{X} \otimes \cdots \otimes \mathcal{X}$, where there are $k$ copies of $\mathcal{X}$.

It is important to observe that in $\mathcal{X} \otimes \mathcal{Y}$, the same state $(s_1, s_2)$ can be reachable through possibly exponentially many paths. This is because the learner may choose to interleave actions in $\mathcal{X}$ with actions in $\mathcal{Y}$ in any order, which means that a state $(s_1, s_2)$ corresponds to exponentially many histories in $\mathcal{X} \otimes \mathcal{Y}$. For that reason, we have to distinguish between *histories* and *states*.

In light of the above discussion, it is inefficient to represent $\mathcal{X} \otimes \mathcal{Y}$ as a tree. Indeed, Zhang et al. [2023] studied *DAG*-form decision problems, and showed that regret minimization on them is possible when the underlying DAG has some natural properties. We state here an immediate consequence of their analysis, which we will use as a black box. Intuitively, the below result states that, as long as utility vectors also only depend on the (terminal) state $\boldsymbol{s}$ that is reached, regret minimization on an arbitrary interleaving of decision problems $\mathcal{X}_1 \otimes \cdots \otimes \mathcal{X}_k$ is possible, and the complexity depends only on the number of states.

**Theorem E.1** (Consequence of Zhang et al., 2023, Corollary A.4). *Let* $\mathcal{X} := \mathcal{X}_1 \otimes \cdots \otimes \mathcal{X}_k$, *where* $\mathcal{X}_i$ *has terminal node set* $\mathcal{Z}_i$. *Let* $\mathcal{Z} := \mathcal{Z}_1 \times \cdots \times \mathcal{Z}_k$ *be the set of terminal states for* $\mathcal{X}$. *For each such terminal state* $z \in \mathcal{Z}$, *let* $V(z)$ *be the set of histories of* $\mathcal{X}$ *whose state is* $z$. *Define the projection* $\pi : \mathcal{X} \to \mathbb{R}^{\mathcal{Z}}$ *by*

$$\pi(\boldsymbol{x})[z] = \sum_{v \in V(z)} \boldsymbol{x}[v].$$

*Then there exists an efficient regret minimizer on* $\pi(\mathcal{X}) := \{\pi(\boldsymbol{x}) : \boldsymbol{x} \in \mathcal{X}\} \subset \mathbb{R}^{\mathcal{Z}}$: *its per-round complexity is* $\mathsf{poly}(|\mathcal{Z}|)$, *and its regret is* $\epsilon$ *after* $\mathsf{poly}(|\mathcal{Z}|)/\epsilon^2$ *rounds.*

Whenever we speak of regret minimizing on interleavings, it will always be the case that utility vectors depend only on the state, so we will always be able to apply the above result. We will call vectors in $\pi(\mathcal{X})$ *reduced strategies*.

Before proceeding, it is instructive to describe in more detail a result of Zhang et al. [2024], which we will also use later, in the language of this section. Let $\mathcal{X}$ and $\mathcal{Y}$ be any two decision problems with terminal node sets $\mathcal{Z}_1$ and $\mathcal{Z}_2$ respectively. A reduced strategy $\boldsymbol{q} \in \pi(\mathcal{X} \otimes \bar{\mathcal{Y}})$ induces a linear map $\phi_{\boldsymbol{q}} : \mathcal{Y} \to \operatorname{conv} \mathcal{X}$, given by

$$\phi_{\boldsymbol{q}}(\boldsymbol{y})[z_1] = \sum_{z_2 \in \mathcal{Z}_2} \boldsymbol{q}[z_1, z_2]\boldsymbol{y}[z_2].$$

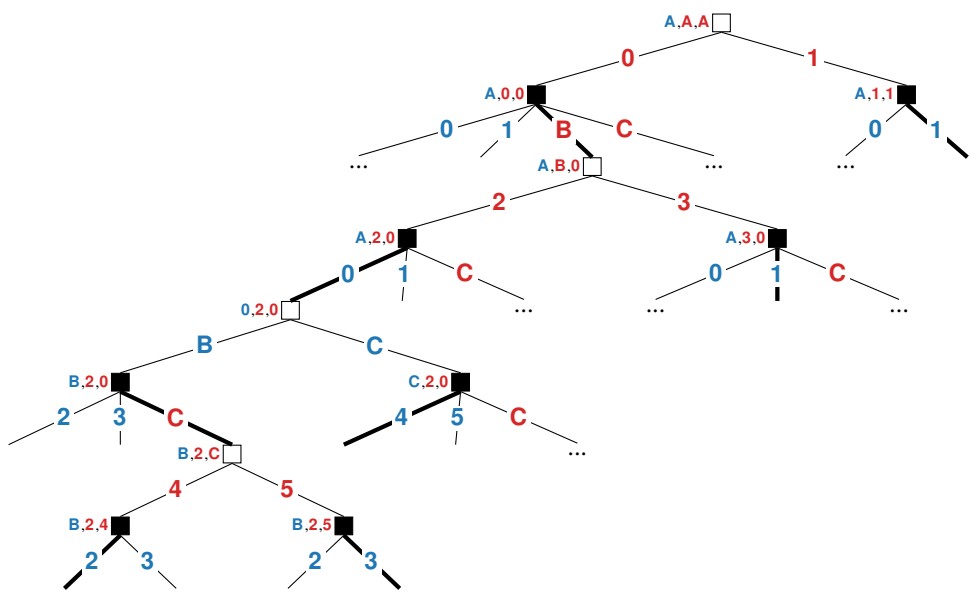

Figure 2: A representation of the deviation $\phi(\boldsymbol{x}) = (x_1 + x_3, x_2x_4, x_2x_5, x_2, 0)$ (discussed in Appendix C.2) in the decision problem $\mathcal{X}$ in Figure 1, as a strategy in $\mathcal{X} \otimes \bar{\mathcal{X}} \otimes \bar{\mathcal{X}}$, *i.e.*, with $k = 2$ mediators. (For an example of a one-mediator deviation, see Zhang et al. [2024, Figure 1].) Again, black squares are decision nodes and white squares are observation nodes. Nodes are labeled with their state representations: the state in $\mathcal{X}$ first (in blue), and the two mediator states after (in red). Similarly, blue edge labels indicate interactions with the decision problem (*i.e.*, playing actions and receiving observations in $\mathcal{X}$), and red edge labels indicate interactions with the mediators (*i.e.*, querying and receiving action recommendations from the mediators). Redundant edges (such as those in which the decision problem in $\mathcal{X}$ has terminated) are omitted. The deviation is shown in thick black lines. For example, $\phi_2(\boldsymbol{x}) = x_2x_4$ because the only state in which the deviator plays action **2** is when the mediator state is (**2**,**4**). $\phi_1(\boldsymbol{x}) = x_1 + x_3$ because the deviator plays action **1** at mediator states (**1**,**1**) and (**3**,**0**), which would give the formula $\phi_1(\boldsymbol{x}) = x_1^2 + x_3x_0$ (where $x_0 := 1 - x_1$), but one can easily check that $x_1^2 + x_3x_0 = x_1 + x_3$ for all $x \in \mathcal{X}$.

It is instructive to think, as Zhang et al. [2024] detailed extensively in their paper, about what strategies $\boldsymbol{q} \in \pi(\mathcal{X} \otimes \bar{\mathcal{Y}})$ represent, and why they induce the linear maps $\phi_{\boldsymbol{q}}$. Decision points $j$ in $\mathcal{Y}$ become observation points in $\mathcal{X} \otimes \bar{\mathcal{Y}}$—at these observation points, the player should observe the action taken by strategy $\boldsymbol{y}$ at $j$. The player in $\mathcal{X} \otimes \bar{\mathcal{Y}}$ is given the ability to *query* the strategy $\boldsymbol{y}$ by *taking the role of the environment* in $\mathcal{Y}$, while the environment, holding a strategy $\boldsymbol{y} \in \mathcal{Y}$, takes the role of the player and answers decision point queries with the actions that it plays. The player then uses these queries to inform how it plays in the true decision problem $\mathcal{X}$. This is the sense in which $\boldsymbol{q}$ induces a map $\phi_{\boldsymbol{q}}$: the output $\phi_{\boldsymbol{q}}(\boldsymbol{y})$ is precisely the strategy that would be played if the environment in $\mathcal{X} \otimes \bar{\mathcal{Y}}$ answers the queries by consulting the strategy $\boldsymbol{y}$. We will call a device that answers queries using strategy $\boldsymbol{y}$ a *mediator holding strategy $\boldsymbol{y}$*. Zhang et al. [2024] then showed the following fact, which we will use critically and repeatedly in the rest of this paper.

**Theorem E.2** (Zhang et al., 2024, Theorem A.2). *Every linear map $\phi : \mathcal{Y} \to \operatorname{conv} \mathcal{X}$ is induced by some reduced strategy $\boldsymbol{q} \in \pi(\mathcal{X} \otimes \bar{\mathcal{Y}})$.*

## E.2 Efficient low-degree swap-regret minimization in extensive-form games

We now proceed with generalizing the results of Appendix D to extensive-form games.

Let $\mathcal{X}$ be any decision problem of dimension $N$ and depth $d$. We will assume WLOG that every decision point in $\mathcal{X}$ has branching factor exactly 2. This is without loss of generality, but it incurs a loss of $O(\log b)$, where $b$ is the original branching factor, in the depth. Thus, in the below bounds, when $d$ appears, it should be read as $O(d \log b)$.

Using the previous notation, the set of *k-mediator deviations* $\Phi_{\text{med}}^k$ is the set of reduced strategies in the decision problem $\mathcal{X} \otimes \bar{\mathcal{X}}^{\otimes k}$. In particular, we recall that reduced strategies $\boldsymbol{q} \in \pi(\mathcal{X} \otimes \bar{\mathcal{X}}^{\otimes k})$ induce functions $\phi_{\boldsymbol{q}} : \mathcal{X} \to \text{conv}\,\mathcal{X}$ given by

$$\phi_{\boldsymbol{q}}(\boldsymbol{x})[z] = \sum_{z_1,\ldots,z_k} \boldsymbol{q}[z, z_1, \ldots, z_k] \prod_{i=1}^k \boldsymbol{x}[z_i].$$

Thus, we have that $\phi_{\boldsymbol{q}}$ is a degree-$k$ polynomial. $\Phi_{\text{med}}^k$ is the set of such deviations. For intuition, we once again pose a few special cases:

- When the original decision problem's decision space is $\Delta_2^N$ (*i.e.*, the decision problem consists of a single root observation point with $N$ children, each of which is a decision point with two actions), we have $\Phi_{\text{DT}}^k = \Phi_{\text{med}}^k$. Thus, the results in this section strictly generalize those in the previous section.

- $\Phi_{\text{med}}^0$ and $\Phi_{\text{med}}^N$ are, as before, the sets of external and swap deviations respectively.

- $\Phi_{\text{med}}^1$ is, by Theorem E.2, the set of all linear deviations.

In this context, applying Theorem E.1 gives an efficient $\Phi_{\text{med}}^k$-regret minimizer:

**Theorem E.3.** *There is an $N^{O(k)}$-time-per-round regret minimizer on $\Phi_{\text{med}}^k$ whose external regret is at most $\epsilon$ after $N^{O(k)}/\epsilon^2$ rounds.*

Thus, once again Proposition C.6 and Theorem C.7 have the following consequence.

**Corollary E.4.** *There is a $N^{O(k)}/\epsilon$-time-per-round regret minimizer on $\mathcal{X}$ whose $\Phi_{\text{med}}^k$-regret is at most $\epsilon$ after $N^{O(k)}/\epsilon^2$ rounds.*

Next, we discuss extensions of our result to low-degree polynomials. Unfortunately, we cannot directly apply Theorem 4.2 to conclude the existence of a regret minimizer on $\mathcal{X}$ with $\Phi_{\text{poly}}^k$-regret growing as $N^{O(k^3)}/\epsilon^2$. There are two issues in attempting to do so.

First, when $\mathcal{X}$ is not the hypercube, polynomials $f : \mathcal{X} \to \{0,1\}$ are not total functions. That is, it is not necessarily the case that degree-$k$ polynomials $f : \mathcal{X} \to \{0,1\}$ can be extended to degree-$k$ polynomials $\bar{f} : \{0,1\}^N \to \{0,1\}$, which is required in order to apply Theorem 4.2.[8] For an example of this, consider $\mathcal{X} = \mathcal{D}_4$ where $\mathcal{D}_N$ is the standard basis in $\mathbb{R}^N$, that is, $\mathcal{D}_N = \{e_i : i \in [N]\}$ where $e_i \in \mathbb{R}^N$ is the $i$th basis vector (in other words, $\mathcal{D}_N$ is the set of vertices of the probability simplex $\Delta(N)$). Let $f : \mathcal{D}_4 \to \{0,1\}$ given by $f(\boldsymbol{x}) = \boldsymbol{x}_1 + \boldsymbol{x}_2$. Then $f$ is linear, but there is no linear $\bar{f} : \{0,1\}^4 \to \{0,1\}$ extending $f$. Indeed, there is a more general manifestation of this phenomenon:

**Proposition E.5.** *For every $N$, there exists a linear map $f : \mathcal{D}_N \to \{0,1\}$ such that any extension $\bar{f} : \{0,1\}^N \to \{0,1\}$ of $f$ must have degree at least $\Omega(\log N)$.*

*Proof.* Let $\bar{f} : \{0,1\}^N \to \{0,1\}$ be any degree-$k$ function. By Theorem 3.4 of O'Donnell [2014], $\bar{f}$ is a $k2^k$-junta, that is, $\bar{f}(\boldsymbol{x})$ depends on at most $k2^k$ entries of $\boldsymbol{x}$. Now consider the map $f : \mathcal{D}_N \to \{0,1\}$ given by $f(\boldsymbol{x}) = \sum_{i \leq N/2} \boldsymbol{x}_i$. Let $\bar{f} : \{0,1\}^N \to \{0,1\}$ be an extension of $f$. Then $\bar{f}$ depends on at least $N/2 - 1$ inputs: if $\bar{f}(\boldsymbol{0}) = 0$ then $\bar{f}$ depends on at least $\boldsymbol{x}_1, \ldots, \boldsymbol{x}_{\lfloor N/2 \rfloor}$, and if $\bar{f}(\boldsymbol{0}) = 1$ then $\bar{f}$ depends on at least $\boldsymbol{x}_{\lfloor N/2 \rfloor+1}, \ldots, \boldsymbol{x}_N$. Thus, we have $N/2 - 1 \leq k2^k$, which upon rearraging gives $k \geq \Omega(\log N)$. $\qquad\square$

The second issue is the following. Suppose that $K$ mediators were enough to represent a function $f : \mathcal{X} \to \{0,1\}$. How does one then represent a function $\phi : \mathcal{X} \to \mathcal{X}$? Each coordinate of $\phi$ could be represented using $K$ mediators, but that need not mean the whole function can. In game-theoretic terms, representing a coordinate of $\phi(\boldsymbol{x})$ allows the player to play a single action, not necessarily the whole game. Naively, playing the whole game would seem to require $Kd$ mediators: $K$ mediators for every level of the decision tree, to compute which action to take at each level.

---

[8]Formally, we call $\bar{f} : \{0,1\}^N \to \{0,1\}$ an *extension* of $f : \mathcal{X} \to \{0,1\}$ if $\bar{f}$ agrees with $f$ on $\mathcal{X}$.

We will show that it is possible to circumvent both of these issues: the first with a loss of $O(d)$ in the degree of the polynomial that is representable, and the second with no additional loss. In particular, we state our main result below.

**Theorem E.6.** $\Phi_{\text{poly}}^k \subseteq \Phi_{\text{med}}^{O(kd)^3}$. *Therefore, for every $k$, there is a $N^{O(kd)^3}/\epsilon$-time-per-round algorithm whose $\Phi_{\text{poly}}^k$-regret at most $\epsilon$ after $N^{O(kd)^3}/\epsilon$ rounds.*

### E.3  Proof of Theorem E.6

We dedicate the rest of this section to the proof of Theorem E.6. We deal with the two aforementioned problems one by one. First, we show that every $f$ admits an extension at a loss of a factor of $d$ in the degree:

**Lemma E.7.** *Every $f : \mathcal{X} \to \{0,1\}$ of degree $k$ admits a degree-$kd$ extension $\bar{f} : \{0,1\}^N \to \{0,1\}$.*

*Proof.* We claim first that the identity function $\text{id} : \mathcal{X} \to \mathcal{X}$ admits a degree-$d$ extension, that is, there is a degree-$d$ function $\bar{\text{id}} : \{0,1\}^N \to \mathcal{X}$ that is the identity on $\mathcal{X}$. Indeed, consider the following function $\bar{\text{id}}$. Then we define $\bar{\text{id}}(\boldsymbol{x})[ja]$ recursively as follows:

$$\bar{\text{id}}(\boldsymbol{x})[ja] = \begin{cases} \boldsymbol{x}[j0] \cdot \bar{\text{id}}(\boldsymbol{x})[p_j] & \text{if} \quad a = 0, \\ (1 - \boldsymbol{x}[j0]) \cdot \bar{\text{id}}(\boldsymbol{x})[p_j] & \text{if} \quad a = 1. \end{cases}$$

It is easy to check that $\bar{\text{id}}$ is indeed the identity on $\mathcal{X}$ by definition of tree-form decision spaces, and that the degree of $\bar{\text{id}}$ is at most the depth of the tree, $d$. But now, for any $f : \mathcal{X} \to \{0,1\}$ of degree $k$, the map $\bar{f} := f \circ \bar{\text{id}} : \{0,1\}^N \to \{0,1\}$ has degree at most $kd$ and is an extension of $f$. $\square$

Now we apply Theorem 4.2. That result tells us that a degree-$kd$ function $\bar{f} : \{0,1\}^N \to \{0,1\}$ can be evaluated using $K = O(kd)^3$ queries. One mediator is certainly more than enough to perform a single query, and therefore such a function can also be evaluated using $K$ mediators. It therefore remains only to address the second problem: namely, the ability to evaluate a function $\bar{\phi} : \{0,1\}^N \to \{0,1\}$, since the output is only binary, in principle only allows us to play a single action. But one needs to play $d$ actions to reach the end of the game. Naively, this would require losing another factor of $d$, for a total of $O(Kd)$ mediators. However, we now show that it is possible to completely circumvent this problem. The notation that we have built up will make this argument perhaps surprisingly short.

Let $\phi : \mathcal{X} \to \mathcal{X}$ be any function such that each component $\phi_z : \mathcal{X} \to \{0,1\}$, given by $\boldsymbol{x} \mapsto \phi(\boldsymbol{x})[z]$, is expressible using $K$ mediators. By definition, $\phi_z$ is expressible as a strategy[9] $\boldsymbol{q}_z \in \pi(\{0,1\} \otimes \bar{\mathcal{X}}^{\otimes K})$. By the argument in the previous section, $\boldsymbol{q}_z$ induces a linear map $\hat{\phi}_z : \overline{X}^{\otimes k} \to \{0,1\}$.

Now let $\hat{\phi} : \overline{\mathcal{X}^{\otimes K}} \to \mathcal{X}$ be the function whose $z$th coordinate is $\hat{\phi}_z$. Every component of $\phi$ is linear, so $\hat{\phi}$ is itself also linear. But then by Theorem E.2, there exists a strategy $\boldsymbol{q} \in \pi(\mathcal{X} \otimes \bar{\mathcal{X}}^{\otimes K})$, that is, a $K$-mediator deviation, that represents $\phi$. This completes the proof.

## F  Discussion and applications

In this section, we discuss various implications and make several remarks about the framework and results that we have introduced.

### F.1  Convergence to correlated equilibria

Notions of $\Phi$-regret correspond naturally to notions of correlated equilibria. Therefore, our results also have implications for no-regret learning algorithms that converge to correlated equilibria. Here, we formalize this connection. Consider an $n$-player game in which player $i$'s strategy set is a tree-form strategy set $\mathcal{X}_i$, and player $i$'s utility is given by a multilinear map $u_i : \mathcal{X}_1 \times \cdots \times \mathcal{X}_n \to$

---

[9]This is a slight abuse of notation since $\{0,1\}$ is not a decision problem, but the argument works if one interprets $\{0,1\}$ as the decision problem with a single decision node and two child terminal nodes. The sense in which $\boldsymbol{q}_z$ represents $\phi_z$ is that, if the environment in $\overline{X}^{\otimes k}$ plays according to a strategy $\boldsymbol{x}$, the player will (eventually) play action $\phi_z(\boldsymbol{x})$.

$[-1, 1]$. For each player $i$, let $\Phi_i \subseteq (\text{conv } \mathcal{X}_i)^{\mathcal{X}_i}$ be a set of deviations for player $i$. Finally let $\Phi = (\Phi_1, \ldots, \Phi_n)$.

**Definition F.1.** A distribution $\pi \in \Delta(\mathcal{X}_1 \times \cdots \times \mathcal{X}_n)$ is called a *correlated profile*. A correlated profile $\pi$ is an $\epsilon$-$\Phi$-*equilibrium* if no player $i$ can profit more than $\epsilon$ via any of the deviations $\phi_i \in \Phi_i$ to its strategy. That is, $\mathbb{E}_{\boldsymbol{x} \sim \pi} u_i(\phi_i(\boldsymbol{x}_i), \boldsymbol{x}_{-i}) \le \mathbb{E}_{\boldsymbol{x} \sim \pi} u_i(\boldsymbol{x}_i, \boldsymbol{x}_{-i}) + \epsilon$ for all players $i$ and $\phi_i \in \Phi_i$.

For example, we can define *k-mediator equilibria* and *degree-k swap equilibria* by setting $\Phi_i$ to $\Phi_{\text{med}}^k$ and $\Phi_{\text{poly}}^k$, respectively. The following celebrated result follows immediately from the definitions of equilibrium and regret.

**Proposition F.2.** *Suppose that every player $i$ plays according to a regret minimizer whose $\Phi_i$-regret is at most $\epsilon$ after $T$ rounds. Let $\pi_i^{(t)} \in \Delta(\mathcal{X}_i)$ be the distribution played by player $i$ at round $t$. Let $\pi^{(t)} \in \Delta(\mathcal{X}_1) \times \cdots \times \Delta(\mathcal{X}_n)$ be the product distribution whose marginal on $\mathcal{X}_i$ is $\pi_i^{(t)}$. Then the average strategy profile, that is, the distribution $\frac{1}{T} \sum_{t \in [T]} \pi^{(t)}$, is an $\epsilon$-$\Phi$-equilibrium.*

Thus, from Theorems E.3 and E.6 it follows, respectively, that, given a game $\Gamma$ where the dimension of each player's decision problem is at most $N$, we have the following results.

**Corollary F.3.** *An $\epsilon$-$k$-mediator equilibrium can be computed in time $N^{O(k)}/\epsilon^3$.*

**Corollary F.4.** *An $\epsilon$-degree-$k$-swap equilibrium can be computed in time $N^{O(kd)^3}/\epsilon^3$.*

The issue of representing the induced correlated distribution is discussed in Appendix G.

### F.2 Strict hierarchy of equilibrium concepts

Let $c \in \{\text{med}, \text{poly}\}$. For every $k \ge 0$, let $\mathcal{E}_c^k(\Gamma)$ be the set of $\Phi_c^k$-equilibria in $\Gamma$. It is clear from definitions that $\mathcal{E}_c^k(\Gamma) \subseteq \mathcal{E}_c^{k-1}(\Gamma)$. Further, even for normal-form games, it is known that coarse-correlated equilibria are not generally equivalent to correlated equilibria, so at least one of these inclusions is strict in some games. We now show that *all* of these inclusions are strict, so that the deviations $\Phi_c^k$ form a *strict* hierarchy of equilibria.[10]

**Proposition F.5.** *For every $k \ge 1$, there exists a game $\Gamma$ such that $\mathcal{E}_c^k(\Gamma) \subsetneq \mathcal{E}_c^{k-1}(\Gamma)$.*

*Proof.* Consider the two-player game $\Gamma$ defined as follows.

- P1's strategy space is $\mathcal{X} = \{-1, 1\}^k$. Player 2's strategy space is simply $\mathcal{Y} = \{-1, 1\}$.[11]

- P1's utility function is $u_1(\boldsymbol{x}, y) = x_1 y$. That is, P1 would like to set $x_1 = y$. P2 gets no utility.

Consider the correlated profile $\pi$ defined as follows: $\pi$ is uniform over the $2^k$ pure profiles $(\boldsymbol{x}, y) \in \mathcal{X} \times \mathcal{Y}$ such that $y = x_1 x_2 \ldots x_k$. P1's expected utility is clearly 0, and there is a swap (*i.e.*, $\Phi_c^k$) deviation that yields a profit of 1, namely $\boldsymbol{x} \mapsto (x_1 x_2 \ldots x_k, \ldots)$. (it does not matter what the swap deviation plays at coordinates other than the first one.) But, since all the $x_i$s are independent, no function of degree less than $k$ can have positive correlation with $x_1 x_2 \ldots x_k$, and thus, there are no profitable deviations of degree less than $k$. Thus, $\pi$ is a $\Phi_c^{k-1}$-equilibrium, but not a $\Phi_c^k$-equilibrium. $\square$

### F.3 Characterization of recent low-swap-regret algorithms in our framework

We have, throughout this paper, introduced and used a framework of $\Phi$-regret that involves fixed points in expectation. Proposition C.3 shows that the ability to compute fixed points in expectation is in some sense necessary for the ability to minimize $\Phi$-regret. It is instructive to briefly discuss how the recent swap-regret-minimizing algorithm of Dagan et al. [2024] and Peng and Rubinstein [2024]

---

[10]The below result constructs a game that depends on $k$. It is *not* the case that there exists a single game for which the inclusion hierarchy is strict: for example, for $k \ge N$, the set $\Phi_c^k$ will already contain all the deviations, so $\mathcal{E}_c^k(\Gamma) = \mathcal{E}_c^N(\Gamma)$ for every $k \ge N$.

[11]These strategy spaces are not technically tree-form strategy spaces, but they are linear transformations of tree-form strategy spaces, so one can also rephrase this argument over tree-form strategy spaces. For cleanliness of notation, we stick to $\{-1, 1\}^k$ as the strategy space.

fits into this framework. Their algorithm makes no explicit reference to fixed-point computation, nor to the minimization of external regret over swap deviations $\phi$—they do not explicitly invoke the framework we use in this paper, nor that of Gordon et al. [2008]. Where is the expected fixed point hidden, then? While we will not present their entire construction here, it suffices to state the following property of it. At every round $t$, the learner outputs a distribution $\pi^{(t)} \in \Delta(\mathcal{X})$ that is uniform on $L$ strategies $\boldsymbol{x}^{(t,1)}, \ldots, \boldsymbol{x}^{(t,L)}$. The way to map this into our framework is to consider $\pi^{(t)}$ an approximate fixed point in expectation of the "function"[12] $\phi^{(t)}$ that maps $\boldsymbol{x}^{(t,\ell)} \mapsto \boldsymbol{x}^{(t,\ell+1)}$ for each $\ell = 1, \ldots, L-1$. With this choice of $\phi^{(t)}$, their algorithm indeed fits into our framework.

### F.4 Revelation principles (or lack thereof)

Most notions of correlated equilibrium obey some form of *revelation principle*. Informally, one can treat a player attempting to deviate profitably from a correlated equilibrium as an interaction between a mediator (who sends useful information to the player) and the player (who tries to play optimally by using the mediator). When studying the regret of online algorithms, one assumes that the interaction with the mediator is *canonical*: the mediator holds with it some sampled strategy profile $(\boldsymbol{x}_1, \ldots, \boldsymbol{x}_n) \sim \pi$, and in equilibrium every player indeed plays $\boldsymbol{x}_i$. We say that the *revelation principle holds* for a particular notion of equilibrium if allowing *non-canonical* equilibria would not expand the set of equilibria. In Appendix H, we give a rather general formalization of this notion, which is enough to encompass all the notions of correlated equilibrium discussed in the paper. We show that, in this formalism, the revelation principle *does not* hold for $k$-mediator equilibria or degree-$k$ swap equilibria when $k > 1$, and indeed in both cases the set of outcomes that can be induced by non-canonical equilibria is the set of *linear-swap* outcomes (Theorems H.4 and H.5).

## G  Representation of strategies

In this section, we discuss how strategies $\pi \in \Delta(\mathcal{X}_1 \times \cdots \times \mathcal{X}_n)$ are represented for the purposes of all of the results in this paper, and in particular for Corollaries F.3 and F.4. In both cases, at each timestep, each player's strategy $\pi_i^{(t)}$ is a uniform mixture of $L = O(1/\epsilon)$ strategies $\delta(\boldsymbol{x}_i^{(t,1)}), \ldots, \delta(\boldsymbol{x}_i^{(t,L)})$, and we have

$$\pi = \frac{1}{T} \sum_{t=1}^{T} \bigotimes_{i=1}^{n} \left( \frac{1}{L} \sum_{\ell=1}^{L} \delta(\boldsymbol{x}_i^{(t,\ell)}) \right), \tag{3}$$

that is, $\pi$ is a uniform mixture of products of mixtures of strategies that are themselves outputs of $\delta$. Thus, if the strategy map $\delta$ is established by convention (for example, as mentioned before, it is often conventional to take $\delta$ to be the behavioral strategy map), it suffices to output $\boldsymbol{x}_i^{(t,\ell)} \in \operatorname{conv} \mathcal{X}_i$ for each $i, t, \ell$.

Suppose that we impose a slightly more stringent restriction on the output format, namely, we want $\pi$ to be a uniform mixture of products of mixtures of *pure* strategies. In that case, we can take $\delta$ to be the Carathéodory map.[13] Now, writing $\boldsymbol{x}_i^{(t,\ell)} = \sum_{j \in [N]} \alpha_i^{(t,\ell,j)} \boldsymbol{x}_i^{(t,\ell,j)}$ for $\boldsymbol{x}_i^{(t,\ell,j)} \in \mathcal{X}_i$, we set

$$\pi = \frac{1}{T} \sum_{t=1}^{T} \bigotimes_{i=1}^{n} \left( \frac{1}{L} \sum_{\ell=1}^{L} \sum_{j=1}^{N} \alpha_i^{(t,\ell,j)} \boldsymbol{x}_i^{(t,\ell,j)} \right). \tag{4}$$

So, our output consists of the strategies $\boldsymbol{x}_i^{(t,\ell,j)} \in \mathcal{X}_i$ and their coefficients $\alpha_i^{(t,\ell,j)}$ for each $i, t, \ell, j$.

## H  On the revelation principle

In this section, we give a formalization of the revelation principle which encompass all the notions of correlated equilibrium discussed in the paper. In this formalism, the revelation principle *does*

---

[12]"Function" is in quotes because the stated $\phi$ may not be a function at all; for example, the sequence $\boldsymbol{x}^{(t,1)}, \ldots, \boldsymbol{x}^{(t,L)}$ may contain repeats yet be aperiodic.

[13]We remind the reader here that the choice of $\delta$ must be the same one that was used to run the algorithm, so by taking $\delta$ here, we mean that we are considering a distribution $\pi$ that was computed by running our algorithm with that choice of $\delta$. That is, the $\pi$ in Eq. (3) is not the same as the $\pi$ in Eq. (4).

*not* hold for $k$-mediator equilibria or degree-$k$ swap equilibria when $k > 1$, and indeed in both cases we show that the set of outcomes that can be induced by non-canonical equilibria is the set of *linear-swap* outcomes (Theorems H.4 and H.5).

Let $\mathscr{D}$ be the class of all finite tree-form decision problems. For each pair $\mathcal{X}, \mathcal{Y} \in \mathscr{D}$ let $\Phi_{\mathcal{X},\mathcal{Y}} \subseteq (\operatorname{conv} \mathcal{X})^{\mathcal{Y}}$ be a subset of deviations. Finally let $\Phi = \bigsqcup_{\mathcal{X},\mathcal{Y} \in \mathscr{D}} \Phi_{\mathcal{X},\mathcal{Y}}$. As in the previous section, consider a game where player $i$ has strategy set $\mathcal{X}_i$ and utility $u_i : \mathcal{X}_1 \times \cdots \times \mathcal{X}_n \to [-1, 1]$.

**Definition H.1.** Let $\mathcal{Y}_1, \ldots, \mathcal{Y}_n \in \mathscr{D}$ be arbitrary tree-form strategy spaces, let $\pi \in \Delta(\mathcal{Y}_1 \times \cdots \times \mathcal{Y}_n)$, and let $\phi_i \in \Phi_{\mathcal{X},\mathcal{Y}}$ for each player $i$. We call the tuple $((\mathcal{Y}_i, \phi_i)_{i=1}^n, \pi)$ a *generalized profile*. A generalized profile is a *generalized $\epsilon$-$\Phi$-equilibrium* if no player $i$ can profit by switching to a different strategy mapping $\phi' : \mathcal{Y}_i \to \operatorname{conv} \mathcal{X}_i$. That is,

$$\mathop{\mathbb{E}}_{\boldsymbol{y} \sim \pi} u_i(\phi'_i(\boldsymbol{y}_i), \phi_{-i}(\boldsymbol{y}_{-i})) \leq \mathop{\mathbb{E}}_{\boldsymbol{y} \sim \pi} u_i(\phi_i(\boldsymbol{y}_i), \phi_{-i}(\boldsymbol{y}_{-i})) + \epsilon$$

for all players $i$ and $\phi'_i \in \Phi_{\mathcal{X},\mathcal{Y}}$. We call a generalized profile *canonical* if $\mathcal{Y}_i = \mathcal{X}_i$ and $\phi_i : \mathcal{X}_i \to \operatorname{conv} \mathcal{X}_i$ is the identity map for every $i$.

In this language, the definitions of equilibrium in Appendix F.1 were definitions of *canonical* equilibria. Every generalized profile induces a canonical profile, namely, the distribution over strategy given by sampling $\boldsymbol{y} \sim \pi$ and returning $(\phi_1(\boldsymbol{x}_1), \ldots, \phi_n(\boldsymbol{x}_n))$. Call two generalized profiles *equivalent* if they induce the same canonical profile. We can now define the revelation principle as follows.

**Definition H.2** (Revelation principle). The class of deviations $\Phi$ *satisfies the revelation principle* if the induced canonical profile of every generalized $\epsilon$-$\Phi$-equilibrium is also an $\epsilon$-$\Phi$-equilibrium.

For an example, let $\Phi$ be the set of all functions, so that the notion of equilibrium is the normal-form correlated equilibrium. Then one can think of the sample $\boldsymbol{y} \sim \pi$ as a profile of signals (one signal per player) from a correlation device, and $\phi_i$ as player $i$'s mapping from signals to strategies. Then the revelation principle states that, without loss of generality (up to utility equivalence), one can assume that signals are recommendations of strategies ($\mathcal{Y}_i = \mathcal{X}_i$) and players in equilibrium play their recommended strategies ($\phi_i : \mathcal{X}_i \to \operatorname{conv} \mathcal{X}_i$ is the identity map).

All notions of equilibrium that we have mentioned can be expressed in this language, and the revelation principle applies to all of them.

**Proposition H.3** (Sufficient conditon for revelation principle). *Let $\delta$ be a consistent strategy map in the sense of Appendix C.2. Suppose that, for every $\phi \in \Phi_{\mathcal{X},\mathcal{Y}}$ and $\psi \in \Phi_{\mathcal{X},\mathcal{X}}$, we have $\psi^\delta \circ \phi \in \Phi_{\mathcal{X},\mathcal{Y}}$. Then $\Phi$ satisfies the revelation principle.*

*Proof.* Given a generalized $\epsilon$-$\Phi$-equilibrium, $((\mathcal{Y}_i, \phi_i)_{i=1}^n, \pi)$, let $\pi'$ be its induced canonical profile. We need to show that $\pi'$ is also an $\epsilon$-$\Phi$-equilibrium. Consider any hypothetical deviation $\psi_i \in \Phi_{\mathcal{X},\mathcal{X}}$. We have

$$\begin{aligned}
\mathop{\mathbb{E}}_{\boldsymbol{x} \sim \pi'} u_i(\psi_i(\boldsymbol{x}_i), \boldsymbol{x}_{-i}) &= \mathop{\mathbb{E}}_{\boldsymbol{y} \sim \pi} u_i((\psi_i^\delta \circ \phi_i)(\boldsymbol{y}_i), \phi_{-i}(\boldsymbol{x}_{-i})) \\
&\leq \max_{\phi_i^* \in \Phi_{\mathcal{X},\mathcal{Y}}} \mathop{\mathbb{E}}_{\boldsymbol{y} \sim \pi} u_i(\phi_i^*(\boldsymbol{y}_i), \phi_{-i}(\boldsymbol{x}_{-i})) \\
&\leq \mathop{\mathbb{E}}_{\boldsymbol{y} \sim \pi} u_i(\phi_i(\boldsymbol{y}_i), \phi_{-i}(\boldsymbol{x}_{-i})) + \epsilon \\
&= \mathop{\mathbb{E}}_{\boldsymbol{x} \sim \pi'} u_i(\boldsymbol{x}_i, \boldsymbol{x}_{-i}),
\end{aligned}$$

where the second line uses the assumed composition property. $\qquad\square$

The revelation principle has, of course, been shown for various special cases of equilibrium before us: for example, NFCE Aumann [1974], linear-swap equilibria Zhang et al. [2024], and so on. Our proposition above generalizes these proofs since each of those notions indeed satisfies the requisite compositional criterion: compositions of arbitrary functions are still arbitrary functions, and compositions of linear maps are linear.

The definition of $k$-mediator functions and degree-$k$ polynomials are both easy to generalize from the $\mathcal{X} \to \mathcal{X}$ case to the $\mathcal{Y} \to \mathcal{X}$ case. We can therefore define *generalized $k$-mediator equilibria* and *generalized degree-$k$ swap equilibria*, and ask whether the revelation principle applies to

these. Proposition H.3 does not apply, because compositions of $k$-mediator and degree-$k$ functions will usually require more mediators and a higher degree. Indeed, we now show that the revelation principle fails for these notions when $k > 1$. We will, in fact, show something quite strong. Recall Proposition F.5, which showed that both canonical $\Phi_{\text{med}}^k$ and canonical $\Phi_{\text{poly}}^k$ are *strictly* tightening notions of equilibrium as $k$ increases. Here we show that this is *not* the case for generalized equilibria.

**Theorem H.4.** *For every $k \geq 1$, every* linear-swap *equilibrium is equivalent to a (non-canonical) $k$-mediator equilibrium.*

*Proof.* Let $\pi$ be any linear-swap equilibrium. We define a mediator of player $i$. Given a strategy $\boldsymbol{x} \in \mathcal{X}$, the mediator will interact with the player as follows. First, for each decision point $j$ of player $i$, and let $a_j \in \mathcal{A}_j$ be the action that is played by $\boldsymbol{x}$. (If $\boldsymbol{x}$ defines no action at $j$, the mediator selects one at random). The mediator samples integers $a_j^{(1)}, \ldots, a_j^{(k)} \in [b_j]$ uniformly at random under the constraint that $\sum_{\ell=1}^k a_j^{(\ell)} = a_j \pmod{b_j}$. The important property that is derived from the above construction is simply that, in order to learn $a_j$, the player must know *all* the $a_j^{(\ell)}$s, and without knowing all of them, the player learns *no information whatsoever*.

When the player arrives, the mediator has the following interaction with the player. First, the player must supply an integer $\ell \in [k]$. We will call $\ell$ the *seed*. Then, whenever the player sends a decision point $j$, the mediator checks whether the decision point sent is consistent with the sequence of decision points previously sent by the player (*i.e.*, if $j$ is a possible next-decision-point following the previous decision point sent). If not, the mediator terminates the interaction. If so, the mediator sends $a_j^{(\ell)}$ to the player.

Now consider how a player can use $k$ copies of such a mediator. In order to learn any useful information at all about any decision point $j$, the player must (1) supply a different seed to each of the $k$ mediators, and (2) query decision point $j$ with all $k$ mediators. Therefore, the player can essentially only use these mediators as if they were one, giving the same sequence of queries to every mediator and computing the true action recommendation by computing $\sum_\ell a_j^{(\ell)} \pmod{b_j}$. Thus, the player can only implement deviations that it could with a single mediator, which by Theorem E.2 are the linear deviations.

What we have thus shown is the following. Let $\mathcal{Y}_i$ be a strategy space that represents the above interaction with the mediator, and let $\phi_i$ be the $k$-mediator deviation that acts according to the previous paragraph. Let $\pi' \in \Delta(\mathcal{Y}_1 \times \cdots \times \mathcal{Y}_n)$ be the distribution in which a strategy $\boldsymbol{x} \sim \pi$ is sampled, the $a_j^{(\ell)}$s are sampled according to the first paragraph, and each mediator plays according to the strategy $\boldsymbol{y}_i$ that answers queries according to those $a_j^{(\ell)}$s. Then $((\mathcal{Y}_i, \phi_i)_{i=1}^n, \pi')$ is a non-canonical $\Phi_{\text{med}}^k$-equilibrium. $\qquad\square$

**Theorem H.5.** *Suppose that every player's strategy space in a given game $\Gamma$ is the hypercube $\mathcal{X} = \{-1, 1\}^N$. Then every linear-swap in $\Gamma$ equilibrium is equivalent to a (non-canonical) degree-$k$-swap equilibrium.*

*Proof.* The proof follows a similar idea to the previous one: we wish to construct a scenario such that, with any polynomial of degree less than $k$, the deviator cannot learn anything about $\boldsymbol{x}$, and with a polynomial of degree $k$ the deviator can only implement linear functions on $\boldsymbol{x}$. Let $\pi$ be any linear-swap equilibrium. Let $\mathcal{Y} = \{-1, 1\}^{Nk}$, and index the coordinates of $\boldsymbol{y} \in \mathcal{Y}$ by pairs $(j, \ell)$ where $j \in [N]$ and $\ell \in [k]$. Define $\phi(\boldsymbol{y})_j = \prod_{\ell \in [k]} \boldsymbol{y}[j, \ell]$. Finally, define the distribution $\pi' \in \Delta(\mathcal{Y}^n)$ as follows: sampling $\boldsymbol{y} \sim \pi'$ is done by sampling $\boldsymbol{x} \sim \pi$, and then, for each player $i$ and index $j$, sampling $\boldsymbol{y}_i[j, \cdot] \in \{-1, 1\}^\ell$ uniformly at random under the constraint that $\prod_\ell \boldsymbol{y}_i[j, \ell] = \boldsymbol{x}_i[j]$. We will write $\pi'|\boldsymbol{x}$ for the conditional distribution of $\boldsymbol{y} \sim \pi'$, conditioned on sampling the given $\boldsymbol{x} \sim \pi$. Now consider the generalized profile $(\mathcal{Y}, \phi, \pi')$ (*i.e.*, every player shares the same signal set $\mathcal{Y}$ and equilibrium deviation function $\phi : \mathcal{Y} \to \mathcal{X}$). We see that it is equivalent to $(\mathcal{X}, \text{id}, \pi)$ by construction. We claim now that it is a degree-$k$ equilibrium which would complete the proof. Consider any degree-$k$ function $\phi' : \mathcal{Y} \to \mathcal{X}$, and define $\psi : \mathcal{X} \to \mathcal{X}$ by $\psi(\boldsymbol{x}) = \mathbb{E}_{\boldsymbol{y} \sim \pi'|\boldsymbol{x}} \phi'(\boldsymbol{y})$. It suffices to show that $\psi$ is a linear map, since then deviating to $\phi'$ would equate to applying the linear

deviation $\psi$ to $\boldsymbol{x}$. Indeed, expressing

$$\phi'(\boldsymbol{y}) = \sum_{S\subseteq[N]\times[k],|S|\leq k} \alpha_S m_S(\boldsymbol{y}) \quad \text{where} \quad m_S(\boldsymbol{y}) := \prod_{(j,\ell)\in S} \boldsymbol{y}[i,s],$$

we see that $\mathbb{E}_{\boldsymbol{y}\sim\pi'|x} m_S(\boldsymbol{y}) = 0$ except when $S = S_j := \{(j,\ell) : \ell \in [k]\}$ for some $j$, in which case $\mathbb{E}_{\boldsymbol{y}\sim\pi'|x} m_S(\boldsymbol{y}) = \boldsymbol{x}[j]$. That is, the only monomials of nonzero expectation are those which multiply together all the $\boldsymbol{y}[j,\cdot]$s, in which case the expectation is exactly $\boldsymbol{x}[j]$. Thus, we have

$$\psi(\boldsymbol{x}) = \sum_S \alpha_S \mathop{\mathbb{E}}_{\boldsymbol{y}\sim\pi'|\boldsymbol{x}}[m_S(\boldsymbol{y})] = \sum_j \alpha_{S_j} \mathop{\mathbb{E}}_{\boldsymbol{y}\sim\pi'|\boldsymbol{x}}[m_{S_j}(\boldsymbol{y})] = \sum_j \alpha_{S_j}\boldsymbol{x}[j]$$

which is indeed linear in $\boldsymbol{x}$. $\qquad\square$

One may wonder at this point about why the above two results do not contradict the fact that NFCE, which supposedly are the $\Phi^N_{\mathrm{med}}$-equilibria, *do* satisfy the revelation principle and are certainly *not* equivalent to linear-swap equilibria. The answer is that the equivalence between swap deviations and $N$-mediator deviations only holds for strategy spaces of size at most $N$—and the strategy spaces $\mathcal{Y}_i$ constructed as part of both proofs have size (at least) $Nk$. When $\mathcal{Y}_i$ has size more than $N$, the set $\Phi^N_{\mathrm{med}}$ is no longer the set of all functions $\mathcal{Y}_i \to \mathrm{conv}\,\mathcal{X}_i$. In other words, the set of *canonical* $\Phi^N_{\mathrm{med}}$-equilibria is equivalent to the set of *canonical* NFCE, but that does not contradict the fact that there exist non-canonical $\Phi^N_{\mathrm{med}}$-equilibria.

# I  Omitted proofs

In this section, we provide the proofs omitted from the previous sections.

## I.1  Proofs from Section B

We first provide the missing proofs from Appendix B. We start with Proposition B.1, the statement of which is recalled below.

**Proposition B.1** (Hazan and Kale, 2007)**.** *Consider a regret minimizer $\mathcal{R}$ operating over $[0,1]^N$. If $\mathcal{R}$ runs in time $\mathrm{poly}(N,1/\epsilon)$ and guarantees $\overline{\mathrm{Reg}}^T_{\Phi^\beta} \leq \epsilon$ for any sequence of utilities, then there is a $\mathrm{poly}(N,1/\epsilon)$ algorithm for computing an $(\epsilon\sqrt{N})$-fixed point of any $\phi^\beta \in \Phi^\beta$ with respect to $\|\cdot\|_2$, assuming that $\phi^\beta$ can be evaluated in polynomial time.*

*Proof.* Suppose that $\mathcal{R}$ outputs a strategy $\boldsymbol{x}^{(t)} \in \mathrm{conv}\,\mathcal{X}$ at each time $t \in [T]$. The basic idea is to determine whether $\|\phi^\beta(\boldsymbol{x}^{(t)}) - \boldsymbol{x}^{(t)}\|_2 \leq \epsilon\sqrt{N}$; if so, the process can terminate as we have identified an $(\epsilon\sqrt{N})$-fixed point of $\phi^\beta$ with respect to $\|\cdot\|_2$. Otherwise, we construct the utility function

$$u^{(t)} : \mathrm{conv}\,\mathcal{X} \ni \boldsymbol{x} \mapsto \frac{1}{\sqrt{N}}\frac{1}{\|\phi^\beta(\boldsymbol{x}^{(t)}) - \boldsymbol{x}^{(t)}\|_2}\langle\phi^\beta(\boldsymbol{x}^{(t)}) - \boldsymbol{x}^{(t)}, \boldsymbol{x} - \boldsymbol{x}^{(t)}\rangle.$$

We note that, by Cauchy-Schwarz, $|u^{(t)}(\boldsymbol{x})| \leq 1$ since $\|\boldsymbol{x} - \boldsymbol{x}^{(t)}\|_2 \leq \sqrt{N}$; hence, the utility function adheres to our normalization constraint. Now, if at all rounds it was the case that $\|\phi^\beta(\boldsymbol{x}^{(t)}) - \boldsymbol{x}^{(t)}\|_2 > \epsilon\sqrt{N}$, we have

$$\overline{\mathrm{Reg}}^T_{\Phi^\beta} \geq \frac{1}{T}\sum_{t=1}^T u^{(t)}(\phi^\beta(\boldsymbol{x}^{(t)})) - \frac{1}{T}\sum_{t=1}^T u^{(t)}(\boldsymbol{x}^{(t)}) > \epsilon, \tag{5}$$

since $u^{(t)}(\boldsymbol{x}^{(t)}) = 0$ and $u^{(t)}(\phi^\beta(\boldsymbol{x}^{(t)})) = \frac{1}{\sqrt{N}}\|\phi^\beta(\boldsymbol{x}^{(t)}) - \boldsymbol{x}^{(t)}\|_2$ for all $t \in [T]$. We conclude that (5) contradicts the assumption that $\overline{\mathrm{Reg}}^T_{\Phi^\beta} \leq \epsilon$ for any sequence of utilities, in turn implying that there exists $t \in [T]$ such that $\|\phi^\beta(\boldsymbol{x}^{(t)}) - \boldsymbol{x}^{(t)}\|_2 \leq \epsilon\sqrt{N}$. Given that we can evaluate $\phi^\beta$ in time $\mathrm{poly}(N)$ (by assumption), we can also compute each error $\|\phi^\beta(\boldsymbol{x}^{(t)}) - \boldsymbol{x}^{(t)}\|_2$ in polynomial time, concluding the proof. $\qquad\square$

We next turn our attention to the proof of Theorem 3.3. We first observe that $\Phi^\beta$ contains functions of the following form.

**Lemma I.1.** $\Phi^\beta$ *contains all functions of the form*

$$\left( \sum_{S_1 \subseteq [N]} \phi_1(S_1) \prod_{j \in S_1} \boldsymbol{x}[j] \prod_{j \in [N] \setminus S_1} (1 - \boldsymbol{x}[j]), \ldots, \sum_{S_N \subseteq [N]} \phi_N(S_N) \prod_{j \in S_N} \boldsymbol{x}[j] \prod_{j \in [N] \setminus S_N} (1 - \boldsymbol{x}[j]) \right),$$

*where $\phi = (\phi_1, \ldots, \phi_N) : \{0,1\}^N \to \{0,1\}^N$. (The convention above is that a product with no terms is to be evaluated as 1.)*

Above, we slightly abuse notation by interpreting $S_j \subseteq [N]$ as a point in $\{0,1\}^N$. Lemma I.1 is evident from the definition of the behavioral strategy map $\beta$: $\phi^\beta(\boldsymbol{x}) = \mathbb{E}_{\boldsymbol{x}' \sim \beta(\boldsymbol{x})} \phi(\boldsymbol{x}')$, and expanding the expectation gives the expression of Lemma I.1—the probability of a set $S$ is exactly $\prod_{j \in S} \boldsymbol{x}[j] \prod_{j \in [N] \setminus S} (1 - \boldsymbol{x}[j])$. We will show that PPAD-hardness for computing fixed points persists under the set of functions contained in $\Phi^\beta$ (per Lemma I.1).

Our starting point is the usual *generalized circuit* problem (abbreviated as GCIRCUIT), introduced by Chen et al. [2009]; it is a generalization of a typical arithmetic circuit but with the twist that it may include *cycles*. (In what follows, we borrow some notation from the paper of Filos-Ratsikas et al. [2023].)

**Definition I.2** (Chen et al., 2009). A generalized circuit with respect to a set of gate-types $\mathcal{G}$ is a list of gates $g_1, \ldots, g_M$. Every gate $g_i$ has a type $G_i \in \mathcal{G}$. Depending on its type, $g_i$ may have zero, one or two input gates, which will be index by $j, k \in [M]$, with the restriction that $i, j, k$ are pairwise distinct.

We denote by $v : [M] \to [0,1]$ a function that assigns an input gate to a value in $[0,1]$. Each gate imposes a constraint induced by its corresponding type. For example, let us consider the following types.

- if $G_i = G_+$, then $v(g_i) = \min(1, v(g_j) + v(g_k))$;
- if $G_i = G_-$, then $v(g_i) = \max(0, v(g_j) - v(g_k))$;
- if $G_i = G_1$, then $v(g_i) = 1$;
- if $G_i = G_{1-}$, then $v(g_i) = 1 - v(g_j)$; and
- if $G_i = G_\times$, then $v(g_i) = v(g_j) \cdot v(g_k)$.

In accordance with the above types, we define $F : [0,1]^M \to [0,1]^M$ to be the function mapping any initial evaluation of the gates to $(v(g_1), \ldots, v(g_M))$. The main problem of interest can be now phrased as follows.

**Definition I.3** ($\epsilon$-GCIRCUIT). The problem $\epsilon$-GCIRCUIT asks for a fixed point of $F$ with respect to $\| \cdot \|_\infty$; that is, an assignment $v : [M] \to [0,1]$ such that all gates are $\epsilon$-satisfied.

A satisfying assignment always exists by Brouwer's theorem, but the associated computational problem is PPAD-hard. In fact, by virtue of a recent result of Filos-Ratsikas et al. [2023], PPAD-hardness persists even if one significantly restricts the type of gates.

**Theorem I.4** (Filos-Ratsikas et al., 2023). *Even when $\mathcal{G} := \{G_+, G_{1-}\}$, there is an absolute constant $\epsilon > 0$ such that computing an $\epsilon$-fixed point of $F$ with respect to $\| \cdot \|_\infty$ is PPAD-complete.*[14]

Yet, the addition gate $G_+$ does not induce a multilinear in the form of Lemma I.1. We will address this by showing that the gate $G_+$ can be approximately simulated via a small number of gates with type either $G_{1-}$ or $G_\times$. To provide better intuition, we first prove this claim under the assumption that the gates are satisfied exactly, and we then proceed with the more general statement. In the sequel, we sometimes use the shorthand notation $t = t' \pm \epsilon \iff t \in [t' - \epsilon, t' + \epsilon]$.

**Lemma I.5.** *Any addition gate $G_+$ can be approximated with at most $\epsilon > 0$ error using $O(\log(1/\epsilon)/\epsilon)$ gates with type either $G_{1-}$ or $G_\times$.*

---

[14]Deligkas et al. [2022] showed that, for a certain variant of this problem, any constant $\epsilon < 0.1$ suffices. It was Rubinstein [2016] who first proved that the problem is PPAD-hard even for a constant $\epsilon > 0$.

*Proof.* Let $t_1^{(1)}, t_2^{(1)} \in [0, 1]$. We define the mapping

$$f : (t_1, t_2) \mapsto (1 - (1 - t_1)(1 - t_2), t_1 t_2).$$

We consider the sequence $t_1^{(\tau+1)}, t_2^{(\tau+1)} := f(t_1^{(\tau)}, t_2^{(\tau)})$ for $\tau = 1, 2, \ldots$. We will show that

$$\min(1, t_1^{(1)} + t_2^{(1)}) = t_1^{(C)} \pm \epsilon, \tag{6}$$

for $C \geq \frac{\log(1/\epsilon)}{\log\left(\frac{1}{1-\epsilon}\right)} + 1 = \Theta(\log(1/\epsilon)/\epsilon)$. We first observe that $t_1^{(\tau+1)} \geq t_1^{(\tau)}$ and $t_1^{(\tau)} + t_2^{(\tau)} = t_1^{(1)} + t_2^{(1)}$. We consider two cases.

- If $t_1^{(C)} > 1 - \epsilon$, then $t_1^{(1)} + t_2^{(1)} = t_1^{(C)} + t_2^{(C)} \geq 1 - \epsilon$, in turn implying that $\min(1, t_1^{(1)} + t_2^{(1)}) \geq 1 - \epsilon$. This clearly implies (6) as $t_1^{(C)} \in [0, 1]$.

- In the contrary case, if $t_1^{(C)} \leq 1 - \epsilon$, then $t_2^{(C)} \leq \prod_{\tau=1}^{C-1} t_1^{(\tau)} \leq (1 - \epsilon)^{C-1} \leq \epsilon$, where we used the fact that $t_1^{(\tau)} \leq t_1^{(C)} \leq 1 - \epsilon$. So, $\min(1, t_1^{(1)} + t_2^{(1)}) = \min(1, t_1^{(C)} + t_2^{(C)}) = t_1^{(C)} + t_2^{(C)} \leq t_1^{(C)} + \epsilon$.

$\square$

**Lemma I.6.** *Any addition gate $G_+$ can be approximated with at most $\epsilon > 0$ error using $O(\log(1/\epsilon)/\epsilon)$ gates with type either $G_{1-}$ or $G_\times$, each with error $\epsilon' = O(\epsilon/C) = O(\epsilon^2)$.*

*Proof.* Let $t_1^{(1)}, t_2^{(1)} \in [0, 1]$. We consider the sequence

$$[0, 1]^2 \ni (t_1^{(\tau+1)}, t_2^{(\tau+1)}) := (1 - (1 - t_1^{(\tau)})(1 - t_2^{(\tau)}) \pm 5\epsilon', t_1^{(\tau)} t_2^{(\tau)} \pm \epsilon') \quad \tau = 1, 2, \ldots.$$

Here, $(t_1^{(\tau+1)}, t_2^{(\tau+1)})$ can be indeed obtained from $(t_1^{(\tau)}, t_2^{(\tau)})$ using 5 gates with type either $G_{1-}$ or $G_\times$, each with error at most $\epsilon'$. We will show that

$$\min(1, t_1^{(1)} + t_2^{(1)}) = t_1^{(C)} \pm \epsilon \tag{7}$$

for $C \geq \frac{\log(6/\epsilon)}{\log\left(\frac{1}{1-\epsilon/12}\right)} + 1 = \Theta(\log(1/\epsilon)/\epsilon)$. Below, we will take $\epsilon' := \epsilon/(12C)$. We first observe that $t_1^{(\tau+1)} + t_2^{(\tau+1)} = t_1^{(\tau)} + t_2^{(\tau)} \pm 6\epsilon'$, thereby implying that $t_1^{(C)} + t_2^{(C)} = t_1^{(1)} + t_2^{(1)} \pm 6C\epsilon'$. Further, $t_1^{(\tau)} \leq t_1^{(\tau+1)} + 5\epsilon'$. Hence, $t_1^{(\tau)} \leq t_1^{(C)} + 5C\epsilon'$. We consider two cases.

- If $t_1^{(C)} > 1 - \epsilon/2$, then $t_1^{(1)} + t_2^{(1)} \geq t_1^{(C)} + t_2^{(C)} - 6C\epsilon' \geq 1 - \epsilon/2 - 6C\epsilon' = 1 - \epsilon$ since $\epsilon' = \epsilon/(12C)$. This implies that $\min(1, t_1^{(1)} + t_2^{(1)}) \geq 1 - \epsilon$, and (7) follows.

- In the contrary case, if $t_1^{(C)} \leq 1 - \epsilon/2$, then $t_2^{(C)} \leq (1 - \epsilon/2 + 5C\epsilon')t_2^{(C-1)} + \epsilon' = (1 - \epsilon/12)t_2^{(C-1)} + \epsilon'$. Thus, $t_2^{(C)} \leq C\epsilon' + (1 - \epsilon/12)^{C-1} \leq \epsilon/4$, where we used the fact that $C \geq \frac{\log(6/\epsilon)}{\log\left(\frac{1}{1-\epsilon/12}\right)} + 1$. In turn, we have $t_1^{(1)} + t_2^{(1)} \leq t_1^{(C)} + t_2^{(C)} + 6C\epsilon' \leq 1 + \epsilon/4$.
  We conclude by observing that $t_1^{(C)} = t_1^{(1)} + t_2^{(1)} - t_2^{(C)} \pm 6C\epsilon' = t_1^{(1)} + t_2^{(1)} \pm 3\epsilon/4$.

$\square$

As a result, for any absolute constant $\epsilon > 0$, we can reduce in polynomial time an $\epsilon$-GCIRCUIT instance consisting of M gates with a set of types $\mathcal{G} = \{G_{1-}, G_+\}$ to an $\epsilon'$-GCIRCUIT instance consisting of $\Theta(\text{M})$ gates with a set of types $\mathcal{G}' = \{G_{1-}, G_\times\}$, so long as $\epsilon' = O(\epsilon^2)$ is sufficiently small. Together with Theorem I.4, we arrive at the following conclusion.

**Proposition I.7.** *Even when $\mathcal{G} := \{G_\times, G_{1-}\}$, there is an absolute constant $\epsilon > 0$ such that computing an $\epsilon$-fixed point of $F$ with respect to $\|\cdot\|_\infty$ is PPAD-complete.*

Having established this reduction, a function $F$ arising from such circuits is clearly a multilinear that can be expressed in the form of Lemma I.1. Indeed, we simply observe that, for $S \subseteq [N]$ and $\bar{S} \subseteq [N]$ with $S \cap \bar{S} = \emptyset$, we have

$$\prod_{j \in S} \boldsymbol{x}[j] \prod_{j \in \bar{S}} (1 - \boldsymbol{x}[j]) = \prod_{j \in [N] \setminus S \cup \bar{S}} (\boldsymbol{x}[j] + 1 - \boldsymbol{x}[j]) \prod_{j \in S} \boldsymbol{x}[j] \prod_{j \in \bar{S}} (1 - \boldsymbol{x}[j])$$

$$= \sum_{S', S''} \prod_{j \in S \cup S'} \boldsymbol{x}[j] \prod_{j \in \bar{S} \cup S''} (1 - \boldsymbol{x}[j]),$$

where the summation is over all partitions $S', S''$ of $[N] \setminus (S \cup \bar{S})$. So, combining with Proposition B.1, we arrive at Theorem 3.3, which is restated below for the convenience of the reader.

**Theorem 3.3.** *If a regret minimizer $\mathcal{R}$ outputs strategies in $[0,1]^N$, it is PPAD-hard to guarantee $\overline{\mathrm{Reg}}_{\Phi^\beta} \leq \epsilon/\sqrt{N}$, even with respect to low-degree deviations and an absolute constant $\epsilon > 0$.*

In the above reduction, we used the inequality $\| \cdot \|_\infty \leq \| \cdot \|_2$ so as to translate the guarantee of Proposition B.1 in the language of Theorem I.4. That inequality can be loose, and so instead let us explain how one can obtain sharper hardness results by relying on a stronger complexity assumption which pertains to the so-called $(\epsilon, \delta)$-GCIRCUIT problem. This relaxes the $\epsilon$-GCIRCUIT problem by allowing at most a $\delta$-fraction of the gates to have an error larger than $\epsilon$. In this context, Babichenko et al. [2016] put forward the following conjecture.

**Conjecture I.8** (Babichenko et al., 2016). *There exist absolute constants $\epsilon, \delta > 0$ such that solving the $(\epsilon, \delta)$-GCIRCUIT problem with M gates requires $2^{\tilde{\Omega}(M)}$ time.*

In light of the simplifications observed by Filos-Ratsikas et al. [2023] (Theorem I.4) in conjunction with Lemma I.6, it is not hard to show that Conjecture I.8 can be equivalently phrased by restricting the gates to involve solely multilinear operations—analogously to Proposition I.7. Namely, if the number of gates increases by a factor of $C = C(\epsilon)$, it suffices to take $\epsilon' = \Theta(\epsilon/C)$ and $\delta' = \Theta(\delta/C)$. Now, the point is that Conjecture I.8 is more aligned with a guarantee in terms of $\| \cdot \|_2$. Indeed, if at least a $\delta$-fraction of the gates incur at least an $\epsilon$ error, it follows that $\|F(\boldsymbol{x}) - \boldsymbol{x}\|_2 \geq \epsilon\sqrt{\delta}\sqrt{N}$ (we can assume here that $\delta N$ is an integer). We can thus strengthen Theorem 3.3 as follows.

**Theorem I.9.** *Suppose that Conjecture I.8 holds. If $\mathcal{R}$ outputs strategies in $[0,1]^N$, guaranteeing $\overline{\mathrm{Reg}}_{\Phi^\beta} \leq \epsilon$ requires time $2^{\tilde{\Omega}(N)}$, even with respect to low-degree deviations and an absolute constant $\epsilon > 0$.*

## I.2 Proofs from Section C

We first spell out the construction that establishes Theorem C.2, which is a refinement of the algorithm of Gordon et al. [2008] described earlier in Appendix A. In particular, the one but crucial difference lies in using an $\epsilon$-expected fixed point (Line 3). In the context of Algorithm 1, we assume that the interface of a regret minimizer $\mathcal{R}$ consists of two components: $\mathcal{R}.\textsc{NextStrategy}()$, which returns the next strategy of $\mathcal{R}$; and $\mathcal{R}.\textsc{ObserveUtility}(\cdot)$, which provides to $\mathcal{R}$ as feedback a utility function, whereupon $\mathcal{R}$ may update its internal state accordingly.

---

**Algorithm 1:** $\Phi$-regret minimizer using fixed points in expectation

---

**Input:** An external regret minimizer $\mathcal{R}_\Phi$ over $\Phi$
**Output:** A $\Phi$-regret minimizer $\mathcal{R}$ over $\mathcal{X}$

1 **function** NEXTSTRATEGY()
2 $\quad$ $\phi^{(t)} \leftarrow \mathcal{R}_\Phi.\textsc{NextStrategy}()$
3 $\quad$ $\pi^{(t)} \leftarrow \epsilon$-expected fixed point of $\phi^{(t)}$ (Definition C.1)
4 $\quad$ **return** $\pi^{(t)}$
5 **function** OBSERVEUTILITY($u^{(t)}$)
6 $\quad$ Set $u_\Phi^{(t)} : \Phi \ni \phi \mapsto \left\langle \boldsymbol{u}^{(t)}, \mathbb{E}_{\boldsymbol{x}^{(t)} \sim \pi^{(t)}} \phi(\boldsymbol{x}^{(t)}) \right\rangle$
7 $\quad$ $\mathcal{R}_\Phi.\textsc{ObserveUtility}(u_\Phi^{(t)})$

---

We next show that there is a certain equivalence between expected fixed points and $\Phi$-regret minimization over $\Delta(\mathcal{X})$, which mirrors the construction of Hazan and Kale [2007].

**Proposition C.3.** *Consider a regret minimizer $\mathcal{R}$ operating over $\Delta(\mathcal{X})$. If $\mathcal{R}$ runs in time* $\mathsf{poly}(N, 1/\epsilon)$ *and guarantees* $\overline{\mathrm{Reg}}_\Phi^T \leq \epsilon$ *for any sequence of utilities, then there is a* $\mathsf{poly}(N, 1/\epsilon)$ *algorithm for computing* $(\epsilon D_\mathcal{X})$*-expected fixed points of* $\phi \in \Phi$, *assuming that we can efficiently compute* $\mathbb{E}_{\boldsymbol{x}^{(t)} \sim \pi^{(t)}}[\phi(\boldsymbol{x}^{(t)}) - \boldsymbol{x}^{(t)}]$ *at any time $t$. Here, $D_\mathcal{X}$ is the diameter of $\mathcal{X}$ with respect to* $\|\cdot\|_2$.

*Proof.* Suppose that $\mathcal{R}$ outputs a strategy $\pi^{(t)} \in \Delta(\mathcal{X})$. At each time $t \in [T]$, we can terminate if $\|\mathbb{E}_{\boldsymbol{x}^{(t)} \sim \pi^{(t)}}[\phi(\boldsymbol{x}^{(t)}) - \boldsymbol{x}^{(t)}]\|_2 \leq \epsilon D_\mathcal{X}$; that is, we have identified an $(\epsilon D_\mathcal{X})$-expected fixed point. Otherwise, we construct the utility function

$$u^{(t)} : \mathcal{X} \ni \boldsymbol{x} \mapsto \frac{1}{D_\mathcal{X}} \frac{1}{\|\mathbb{E}_{\boldsymbol{x}^{(t)} \sim \pi^{(t)}}[\phi(\boldsymbol{x}^{(t)}) - \boldsymbol{x}^{(t)}]\|_2} \left\langle \mathbb{E}_{\boldsymbol{x}^{(t)} \sim \pi^{(t)}}[\phi(\boldsymbol{x}^{(t)}) - \boldsymbol{x}^{(t)}], \boldsymbol{x} - \pi^{(t)} \right\rangle,$$

which indeed satisfies the normalization constraint $|u^{(t)}(\boldsymbol{x})| \leq 1$. Now, if at all iterations it was the case that $\|\mathbb{E}_{\boldsymbol{x}^{(t)} \sim \pi^{(t)}}[\phi(\boldsymbol{x}^{(t)}) - \boldsymbol{x}^{(t)}]\|_2 > \epsilon D_\mathcal{X}$, we have

$$\overline{\mathrm{Reg}}_\Phi^T \geq \frac{1}{T} \sum_{t=1}^T u^{(t)} \left( \mathbb{E}_{\boldsymbol{x}^{(t)} \sim \pi^{(t)}}[\phi(\boldsymbol{x}^{(t)})] \right) - \frac{1}{T} \sum_{t=1}^T u^{(t)}(\pi^{(t)}) > \epsilon$$

since $u^{(t)}(\pi^{(t)}) = 0$ and

$$u^{(t)} \left( \mathbb{E}_{\boldsymbol{x}^{(t)} \sim \pi^{(t)}}[\phi(\boldsymbol{x}^{(t)})] \right) = \frac{1}{D_\mathcal{X}} \left\| \mathbb{E}_{\boldsymbol{x}^{(t)} \sim \pi^{(t)}}[\phi(\boldsymbol{x}^{(t)}) - \boldsymbol{x}^{(t)}] \right\|_2$$

for all $t \in [T]$. This contradicts the assumption that $\overline{\mathrm{Reg}}_\Phi^T \leq \epsilon$ for any sequence of utilities, in turn implying that there exists $t \in [T]$ such that $\pi^{(t)}$ is an $\epsilon$-expected fixed point. Given that, by assumption, we can compute $\mathbb{E}_{\boldsymbol{x}^{(t)} \sim \pi^{(t)}}[\phi(\boldsymbol{x}^{(t)}) - \boldsymbol{x}^{(t)}]$ for any time $t$, the claim follows. $\square$

### I.3 Proof of Corollary 4.1

In this subsection, we discuss how expected fixed points (per Definition C.1) can be used to speed up equilibrium computation even in settings where actual fixed points can be computed in polynomial time. In particular, we recall the following result, which was stated earlier in the main body.

**Corollary 4.1.** *For any $n$-player game in normal form, there is an algorithm that computes an $\epsilon$-correlated equilibrium and runs in time*

$$O\left( \frac{A \log A}{\epsilon^2} \left( \mathsf{EO}(n, A) + n \frac{A^2}{\epsilon} \right) \right).$$

Indeed, using the algorithm of Blum and Mansour [2007] instantiated with MWU,[15] one can guarantee (average) swap regret bounded as $O(\sqrt{A \log A / T})$, where $T$ is the number of iterations; hence, taking $T = O(A \log A / \epsilon^2)$ guarantees at most $\epsilon$ swap regret for each player. Further, a function here can be represented as a stochastic matrix on $A$ states; as such, it can be evaluated at any point in time $O(A^2)$ via a matrix-vector product. As a result, each iteration of Algorithm 1 can be implemented with a single oracle call to (2), along with $n$ updates—one for each player—each of which has complexity bounded as $O(A^2/\epsilon)$ (Theorem C.7). This implies Corollary 4.1.

Let us compare the complexity of Corollary 4.1 with prior results. First, when $\epsilon \gg 0$ the best running time follows from the recent algorithms of Dagan et al. [2024] and Peng and Rubinstein [2024], but those quickly became superpolynomial even when $\epsilon \approx 1/\log A$; indeed, the number of iterations of their algorithm scales as $(\log A)^{\tilde{O}(1/\epsilon)}$. On the other end of the spectrum, one can compute an exact correlated equilibrium in polynomial time using the "ellipsoid against hope" algorithm [Papadimitriou and Roughgarden, 2008, Jiang and Leyton-Brown, 2011]. The original paper by Papadimitriou and Roughgarden [2008] did not specify the exact complexity of the algorithm, but the subsequent work of Jiang and Leyton-Brown [2011]—which analyzed a slightly different

---

[15]The exponential weights map can generate irrational outputs, but this can be addressed by truncating to a sufficiently large number of bits, which does not essentially affect the regret.

version based on a derandomized separation oracle—came up with a bound of $n^6 A^{12}$ in terms of the number of iterations of the ellipsoid; the overall running time is higher than that. While the analysis of Jiang and Leyton-Brown [2011] can likely be significantly improved—as acknowledged by the authors, we do not expect that algorithm to be competitive unless one is searching for very high-precision solutions.

The original algorithm of Blum and Mansour [2007]—instantiated as usual with `MWU`—requires computing a stationary distribution of a Markov chain in every iteration. This amounts to solving a linear system, which in theory requires a running time of $O(A^\omega)$, where $\omega \approx 2.37$ is the exponent of matrix multiplication [Williams et al., 2024]. Without fast matrix multiplication—which is currently widely impractical—the running time would instead be $O(A^3)$. A strict improvement over that running time can be obtained using the *optimistic* counterpart of `MWU` (`OMWU`) [Daskalakis et al., 2021], which has been shown to reduce the number of iterations to $\tilde{O}(A/\epsilon)$ without essentially affecting the per-iteration complexity. Corollary 4.1 improves over that in the regime where $\epsilon \geq 1/A^{\frac{\omega}{2}-1}$, where $\omega \approx 2.37$ is the exponent of matrix multiplication [Williams et al., 2024]; without fast matrix multiplication, the lower bound is instead $\epsilon \geq 1/\sqrt{A}$.

Another notable attempt at improving the complexity of Blum and Mansour [2007] was made by Greenwald et al. [2008] using power iteration. For a parameter $p \geq 1$, their algorithm guarantees at most $O(\sqrt{Ap/T})$ average internal regret with a per-iteration complexity of $\Omega(A^3/p)$ (without fast matrix multiplication). Casting this guarantee in terms of swap regret, and observing that the overall running time is invariant on $p$, we see that the resulting complexity is no better than that via `OMWU`, which we discussed earlier; the approach of Greenwald et al. [2008] is based on regret matching, which incurs an inferior regret bound compared to `MWU`, but is nevertheless known to perform well in practice. An improvement to the algorithm of Blum and Mansour [2007] was obtained by Yang and Mohri [2017], but requires a further assumption on the minimum probability given to every expert; it is unclear whether such an assumption can be guaranteed in general.

The above discussion concerns algorithms operating with full feedback. In the bandit feedback, Ito [2020] came up with a way to update a single column in each iteration, but it still requires computing a stationary distribution in every iteration. Ito [2020] claims that this can be achieved in time almost quadratic time through the work of Cohen et al. [2017]; however, the complexity of the algorithm of Cohen et al. [2017] depends on the condition number of the Markov chain, and it is unclear how to bound that in our setting. On the other hand, algorithms in the bandit setting do not require access to an expectation oracle. This can be beneficial in certain settings, but our discussion here focuses on the regime where the expectation oracle does not dominate the per-iteration complexity. We conjecture that Theorem C.7 can be generalized to the bandit setting, in which case our improvement will also manifest in the regime where the cost of the expectation oracle far outweighs the per-iteration complexity of Theorem C.7, but that is not within our scope in this paper.

