# OpenReview forum: "Efficient $\Phi$-Regret Minimization with Low-Degree Swap Deviations in Extensive-Form Games"
_NeurIPS.cc/2024/Conference — NeurIPS 2024 poster_

### Official Review · Reviewer_o8Ue · 2024-07-05

**Soundness:** 3
**Presentation:** 2
**Contribution:** 3
**Rating:** 6
**Confidence:** 1

**Summary:**

This paper explores and advances the research direction of linear-swap regret in extensive-form games, extending it to low-degree swaps and bridging the gap to results on general swap regret. The technical contributions include the concept of k-mediator deviations, which relate to low-degree polynomials and depth-k decision trees on n>>k variables, and a crucial observation that if a player is allowed to output a probability distribution over strategies, computing it is sufficient to compute an approximate fixed point of the deviations in expectation.

The concept of depth-k deviations allows a deviator to select an action after observing a strategy on a single terminal for k adaptive rounds. Choosing $k=1$ or $k=N$ leads to linear-swap deviations or all swap deviations, respectively. The main results are as follows:

1. For depth-k deviations and the strategy space of a hypercube, there is an online learning algorithm that achieves at most $\\epsilon$ average $\\Phi$-regret in $N^{O(k)} / \\epsilon^2$ rounds with $N^{O(k)} / \\epsilon$ running time per round.


1. For deviations that are polynomials of degree k and the strategy space of a hypercube, there is an online learning algorithm that achieves at most $\\epsilon$ average $\\Phi$-regret in $N^{O(k)} / \\epsilon^2$ rounds with $N^{O(k^3)} / \\epsilon$ running time per round.
2. It is PPAD-hard to guarante a regret of at most $\\epsilon / \\sqrt{N}$ if a regret minimizer outputs a single strategy in $\[0,1\]^N$.

3. For depth-k deviations and extensive-form games, there is an online learning algorithm that achieves at most $\\epsilon$ average $\\Phi$-regret in $N^{O(k)} / \\epsilon^2$ rounds with $N^{O(k)} / \\epsilon$ running time per round.

4. For deviations that are polynomials of degree k and extensive-form games of depth $d$, there is an online learning algorithm that achieves at most $\\epsilon \\Phi$ average $\\Phi$-regret in $N^{O(kd)^3} / \\epsilon^2$ rounds with $N^{O(kd)^3} / \\epsilon$ running time per round.

**Strengths:**

- The results bridge the gap between linear swap and arbitrary swap regret.
- The technical contributions show the advantage of probabilistic strategies with respect to hardness, i.e., that a fixed point in expectation is enough for optimization.

**Weaknesses:**

- It seems the gap is bridged "from below", i.e., for large $k$ the results may become worse than known bounds.
- ~~Allowing the learner to output probability distributions of strategies seems to weaken the model and/or the guarantees of results in this model, e.g., more than allowing mixed strategies over pure strategies does.~~ (edit: see rebuttal)

**Questions:**

Could you elaborate on the difference between mixed strategies and probability distributions over strategies? Is it true that the worst-case regret for the former is bounded, but (at least in your results) the worst-case regret is unbounded?

Note: As a reader who is completely unfamiliar with the topic, the preliminaries could have covered a *deviations*.

**Limitations:**

/

---

> ### Author Rebuttal · Authors · 2024-08-07
>
> *It seems the gap is bridged "from below", i.e., for large $k$ the results may become worse than known bounds.*
>
> This is correct, in particular, for $k > \tilde O(1/\epsilon)$ our bounds become worse than those of Peng and Rubinstein [2024] and Dagan et al [2024].
>
> *Allowing the learner to output probability distributions of strategies seems to weaken the model and/or the guarantees of results in this model, e.g., more than allowing mixed strategies over pure strategies does.*
>
> We do not believe that working with mixed rather than behavioral strategies (see also next response regarding the difference between mixed and behavioral) is "weakening" in any meaningful sense. Indeed:
> * In Theorem 3.3, we show that if a learner is restricted to playing only behavioral strategies, it is PPAD-hard to achieve $\Phi$-regret $\epsilon/\sqrt{N}$ even when $\Phi$ is the set of degree-$k$ polynomials and $k$ and $\epsilon$ are absolute constants. Thus, in some sense, going beyond behavioral strategies to use mixed strategies is *necessary* to achieve efficient learning algorithms.
> * Peng and Rubinstein [2024] and Dagan et al [2024] also use mixed rather than behavioral strategies to achieve their result (this should be unsurprising, given the previous bullet!)
>
> *Could you elaborate on the difference between mixed strategies and probability distributions over strategies?*
>
> A mixed strategy *is* a probability distribution over pure strategies. The reviewer might be confusing a mixed strategy with a *behavioral* strategy. A behavioral strategy is a mixed strategy that selects actions independently at every decision point. Each point $\boldsymbol x \in \text{conv}(\mathcal X)$ essentially uniquely\* specifies a behavioral strategy, but does not uniquely specify a mixed strategy.
>
> \*except at decision points reached with probability zero
>
> *Is it true that the worst-case regret for the former is bounded, but (at least in your results) the worst-case regret is unbounded?*
>
> As stated above, Theorem 3.3 shows that no efficient regret minimizer can exist that only uses behavioral strategies, unless P = PPAD. Theorems 3.1, 3.2, and 3.4 show that, when mixed strategies are permitted, efficient regret minimizers *do* exist.

---

> > ### Comment · Reviewer_o8Ue · 2024-08-08
> >
> > Thank you for the response and the clarifications! Based on your explanations and pointing out that I was at least partially confused about different types of strategies, I withdraw the second weakness. Given that my rating is an educated guess because of my lack of expertise, it affects my rating slightly positively, but in a way that's not mirrored by the rating scale.

---

### Official Review · Reviewer_aMpb · 2024-07-11

**Soundness:** 3
**Presentation:** 2
**Contribution:** 2
**Rating:** 5
**Confidence:** 4

**Summary:**

This work aims to bridge the gap between the $N^{O(1/\varepsilon)}$ result for attaining the $\varepsilon$ swap regret and the $\operatorname{poly}(N)/\varepsilon^2$ result for attaining the $\varepsilon$ linear-swap regret for extensive-form games. To this end, the authors generalize the untimed communication deviations proposed by [1] to k-mediator deviations. The authors prove that obtaining the $\varepsilon$ regret w.r.t. the set of k-mediator deviations in $N^{O(k)}/\varepsilon^2$ rounds is achievable. As several byproducts, the authors show a sample complexity of $N^{O(kd)^3}/\varepsilon^2$ for minimizing the regret against degree-k polynomial deviations, and an algorithm with the lowest computation complexity to compute the $\varepsilon$-correlated equilibria in normal-form games in the medium-precision regime.

[1] Zhang et al. Mediator Interpretation and Faster Learning Algorithms for Linear Correlated Equilibria in General Extensive-Form Games. ICLR, 24.

**Strengths:**

1.	The motivation to bridge the gap between the $N^{O(1/\varepsilon)}$ result for attaining the $\varepsilon$ swap regret and the $\operatorname{poly}(N)/\varepsilon^2$ result for attaining the $\varepsilon$ linear-swap regret is of interest.

**Weaknesses:**

My main concern regarding this work is the presentation. Several key parts of the current writing are not very clear and somewhat hard to follow, detailed as follows.

1.	The main motivation of this work is to bridge the gap between the $N^{O(1/\varepsilon)}$ result for attaining the $\varepsilon$ swap regret and the $\operatorname{poly}(N)/\varepsilon^2$ result for attaining $\varepsilon$ linear-swap regret in previous works. Does the current result of the $N^{O(k)}/\varepsilon^2$ sample complexity for obtaining the $\varepsilon$ regret w.r.t. the set of k-mediator deviations well interpolate between the above two results? It is a bit strange that there seem to be no discussions about the relationship between the results in this work and the two results in previous works. I think the result in this work replicates the $\operatorname{poly}(N)/\varepsilon^2$ result when $k=1$, but it is unclear to me in which sense the result in this work will replicate the previous $N^{O(1/\varepsilon)}$ result.
2.	Why do we need to relate the k-mediator deviations to low-degree polynomials? After relating the k-mediator deviations to low-degree polynomials, the convergence rate is worse by a $N^{O(k^2d^3)}$ factor. There might be some benefits in doing so, but I did not seem to find the relevant descriptions in the paper.
3.	There is a rich line of most related works in the literature, while it is currently a bit heavy to fully understand the advantages of the results in this work against those in previous works merely from the statements in Section 5. It would be better if a table reflecting the convergence rate, the computation complexity, and the required regime of $\varepsilon$ of the results in this work and those in previous works is included in the paper.
4.	There is an expectation taken over the randomness of the player in the regret defined in Eq. (1), while the previous works study high probability regret (say [2,3]). As such, it seems that the algorithm in this work can only obtain approximate correlated equilibria in expectation, while previous works can obtain the approximate correlated equilibria with high probability. I would suggest the authors incorporate necessary discussions about this point.
5.	For Section 4.1, it seems that the proposed operation of approximating fixed points in expectation can only work with finding approximate correlated equilibria in expectation. Can this trick be extended to the case of finding approximate correlated equilibria with high probability? This is also somewhat related to my question above.
6.	In Line 135, it is not clear to me what is the purpose of constructing the depth-k decision tree deviations. It would be better to first introduce the motivation for doing so before diving into the details of its construction.
7.	At the end of the statement of Theorem 3.4, it seems that it should be something like $N^{O(k)}/\varepsilon$ (or $N^{O(k)}/\varepsilon^2$) instead of $N^{O(k)}$?
8.	In Line 228, what does it mean by “succinct representation”?
9.	For Section 4.2, my understanding is that the authors consider a more convenient and suitable formulation than the tree-form formulation to better leverage the connection between low-depth decision trees and low-degree polynomials. Are there any technical difficulties in achieving this?
10.	In this work, most of the time $\Phi$ is termed as the set of the deviations while sometimes it is termed as the set of the transformation functions. I would suggest the authors unify the descriptions for clarity.

Overall, the current writing prevents me from well evaluating the significance of the results and the technical novelty of this work and hence I am not able to recommend an acceptance of this work.

[2] Bai et al. Efficient Phi-Regret Minimization in Extensive-Form Games via Online Mirror Descent. NeurIPS, 22.

[3] Farina et al. Polynomial-Time Linear-Swap Regret Minimization in Imperfect-Information Sequential Games. NeurIPS, 23.

**Questions:**

Please see the weakness part above.

**Limitations:**

Not applicable.

---

> ### Author Rebuttal · Authors · 2024-08-07
>
> 1. *On "Bridging the gap"*: You are correct: while we match the $\text{poly}(N)/\epsilon^2$ result for linear-swap regret when $k=1$, we do not match the $N^{\tilde O(1/\epsilon)}$ bound for swap regret shown by Dagan et al. and Peng and Rubinstein.
> 2. *$k$-mediator deviations and low-degree polynomials*: Note that the $k$ in the $N^{O(k)}/\epsilon^2$ bound refers to something different than the $k$ in the $N^{O(k^3)}/\epsilon^2$ bound: the former is the number of mediators; the latter is the degree of a polynomial. As all $k$-mediator deviations are degree-$k$ polynomials (but not vice-versa), it should not be surprising that our bound in the latter case is worse.
> 3. Thanks for the suggestion. We have created a table comparing the results in that section to past results, and attached it in our message to all reviewers. We will include this table in the next version.
> 4. A clarification here: Our algorithms are *deterministic*--they produce as their output on each round $t$ a *distribution* $\pi^t$. The fact that Eq. (1) contains an expectation simply is how regret is defined, and does not denote that the algorithm ever actually samples from $\pi^t$. We will makes this more clear in the revised version as we see that it can cause confusion.
> 5. Like above, a "fixed point in expectation" is actually a *distribution* $\pi^t$, which is computed by a deterministic algorithm.
> 6. There are (at least) two reasons to consider the set of depth-$k$ decision trees: first, they are by themselves a fairly natural class of functions, and second, our result about low-degree deviations fundamentally uses a connection between degree-$k$ polynomials and depth-$O(k^3)$ decision trees (Theorem 4.2). We will add a note about this to the next version.
> 7. That sentence should read "For $\Phi^k_\text{med}$, the same bounds hold except that both instances of $N^{O(kd)^3}$ should be replaced by $N^{O(k)}$." We will fix this in the next version.
> 8. "Succinct" here simply means the game is given by an efficient oracle for computing the utility vectors (Eq. (2)), rather than as an explicit payoff tensor. This is fairly standard language, also used by (among many other papers) the cited paper [Papadimitriou and Roughgarden 2008]
> 9. In the first part of Section 4.2, to provide better intuition about the general case, we indeed focus on a very specific type of tree-form decision problems (corresponding to the hypercube). As the reviewer points out, this is convenient because of the connection between low-depth decision trees and low-degree polynomials (Theorem 4.2). The main technical challenge is to extend this to general tree-form decision problems, and that is the subject of the rest of Section 4.2.
> 11. Thank you. As you correctly point out, "deviation" and "[strategy] transformation function" are the same thing. We'll unify the terminology in the next version.

---

> > ### Comment · Reviewer_aMpb · 2024-08-12
> >
> > Thanks for your responses. However, I am still not fully convinced by your responses to Q4 and Q5. Why is your algorithm deterministic? Since in each round, your algorithm will actually sample an action $x^t\sim \pi^t$, it can hardly be said that your algorithm is deterministic unless all the $\pi^t$'s are Dirac measures over the action space. Further, even if your algorithm is deterministic, why could you guarantee a sublinear regret result against the adversary? It is known that a deterministic algorithm can have linear regret against adversarial multi-armed bandits [4].
> >
> > [4] Lattimore et al. Bandit Algorithms. 2020.

---

> > > ### Author Response · Authors · 2024-08-12
> > >
> > > Thank you for the response. Our algorithm is indeed deterministic. We never sample an action from $\pi^t$. The output of the learner is that distribution instead of a sample from it. This is similar to Hedge where the output is a distribution over actions, as opposed to EXP3 where the output is a sample from the distribution. Using deterministic algorithms is in line with much of the prior work on the subject; see, for example, [1,2,3] and references therein. Also, note that we operate in the full feedback model, and not under bandit feedback (see Lines 105-107 in our paper). In that setting, there are many well-known deterministic algorithms with sublinear regret, such as online gradient descent and Hedge, so the lower bound cited by the reviewer is not applicable.
> > >
> > > We hope this clarifies the reviewer’s question, and we will highlight further in the revision that no randomization is used in our algorithm.
> > >
> > > [1] Piliouras et al. Beyond Time-Average Convergence: Near-Optimal Uncoupled Online Learning via Clairvoyant Multiplicative Weights Update, NeurIPS 2022
> > >
> > > [2] Daskalakis et al. Near-Optimal No-Regret Learning in General Games, NeurIPS 2021
> > >
> > > [3] Syrgkanis et al. Faster Convergence of Regularized Learning in Games, NIPS 2015

---

> > > > ### Comment · Reviewer_aMpb · 2024-08-12
> > > >
> > > > Thank you for the replies. Most of my concerns have been solved. I now believe the results of this work might be of interest to the community. However, as noted before, some parts of the presentation of this work are not clear enough and I believe it might need an in-depth revision in the future for better clarity. As such, I have marginally increased my score to 5.

---

### Official Review · Reviewer_VzB7 · 2024-07-15

**Soundness:** 3
**Presentation:** 2
**Contribution:** 3
**Rating:** 6
**Confidence:** 1

**Summary:**

This work seems to design some fast regret minimization algorithms. However, to be honest, I could not find the learning protocols nor understand the learning problem set-up. It is hard for me to find what information is revealed in each round after taking some action.
So, I suggest adding some simple applications to help readers to understand the learning protocol, for example, the tree-form decision problems.

Major comments:

Is there any lower bound? How can I measure the tightness of the presented results?
Is it related to P vs NP?  Can it be reduced to an NP-hard problem?
For theoretical analysis, is it finite-time analysis or asymptotic analysis?
I know it is a theory work, but is it any experiments?

Minor comments:

In Line 110, a comma is missing in $\{ \ \}$.

In Line 143, it is better to say $x[j]'s$.



In Line 271, it should be \citep{} for the citation.

**Strengths:**

see the first box

**Weaknesses:**

see the first box

**Questions:**

see the first box

**Limitations:**

I hope to see some experiments though

---

> ### Author Rebuttal · Authors · 2024-08-07
>
> Q: *However, to be honest, I could not find the learning protocols nor understand the learning problem set-up.*
>
> A: We will clarify further how the learning protocol proceeds in the revised version. Note that we operate in the standard model when it comes to learning in extensive-form games.
>
>
> Q: *Is there any lower bound? How can I measure the tightness of the presented results? Is it related to P vs NP? Can it be reduced to an NP-hard problem?*
>
> A: The paper [Anonymous, 2024] that was included in the supplemental materials shows an exponential lower bound in the case of *swap regret*, that is, when $k=N$. One can interpret our results as "interpolating" between the known extreme cases of $k=1$ (where efficient algorithms exist due to Farina & Pipis [2023] and Zhang et al [2024]) and $k=N$, where the above lower bound holds. See also response to aMpb's Q1 below.
>
>
> Q: *For theoretical analysis, is it finite-time analysis or asymptotic analysis?*
>
> A: The theoretical analysis is all finite-time. Each theorem specifies an upper bound on the number of rounds required to achieve a certain equilibrium gap $\epsilon$. The big-O notation hides only absolute constants.
>
> Q: *I know it is a theory work, but is it any experiments?*
>
> A: We agree that the main merit of the paper is theoretical. Our notion of rationality is the strongest that can be efficiently (i.e., at a polynomially sublinear regret rate) guaranteed in imperfect-information sequential games (i.e., extensive-form games), a fundamental class of strategic interactions. We have not run experiements using our algorithm, but we agree that this may be something interesting to do in the future.
>
> Thank you for pointing out the minor errors. We will fix them in the next version.

---

### Author Rebuttal · Authors · 2024-08-07

Thanks for the reviews! We have attached to this message a pdf containing two tables which compare the results of this paper to results of past papers. We will include both of these tables in the next version.

---

### Decision · Program_Chairs · 2024-09-25

**Decision:**

Accept (poster)

**Comment:**

The submission gives a (slow but) polynomial time algorithm for minimizing \Phi-regret wrt to a few classes of deviations such as low-degree swaps. This generalizes previous work on linear deviations.

The reviews and discussion were inconclusive, so I also read the paper myself.
Some points:

1. Two reviewers (and I, independently) were annoyed with the claim in the abstract that in this paper "take[s] a step toward bridging the gap between those two results". The bounds in the paper deteriorate quickly when $k$ is large.

2. The authors don't bother motivating the classes of deviations that they consider. I feel like this is a major drawback.

3. I really like the separation behavioral and more general mixed strategies! I feel like this a nice and somewhat surprising contribution.

4. There were some concerns in reviews about presentation.


Overall I'm leaning toward accepting the paper due to #3. This is in hope that the authors address #4, and soften the abstract to address #1.